# Transiently delocalised hybrid quantum states are gateways for efficient exciton dissociation at organic donor-acceptor interfaces

Filip Ivanović[1], Samuele Giannini [2], Wei-Tao Peng[3] & Jochen Blumberger [1] ✉

The field of organic photovoltaics has witnessed a renaissance in recent years owing to the development of non-fullerene acceptor materials reaching record power conversion efficiencies of > 20%. New computational models are needed to rationalise the regimes of photophysics reached in these materials. Here we report on a novel implementation of eXcitonic state-based Surface Hopping, a powerful non-adiabatic molecular dynamics method for the simulation of photo-induced charge generation in truly nanoscale donor-acceptor interfaces. We observe a transition from an inefficient cold to an efficient hot exciton dissociation mechanism as the electronic coupling between the molecules or the dielectric constant of the materials is increased. The hot pathway is observed to occur by Frenkel excitons converting into transiently delocalised hybrid exciton-charge transfer states that subsequently form charge separated states. This avoids the formation of kinetically trapped interfacial charge transfer states that are prone to non-radiative recombination.

Organic solar cells (OSCs) have attracted considerable research attention in recent decades, largely due to their relatively low processing costs and mechanical flexibility allowing applications that cannot be accessed using inorganic materials[1]. The introduction of a two-component OSC or heterojunction, where donor and acceptor phase are separated by an interface, constituted a seminal leap towards the commercial application of OSCs[2], whilst the introduction of non-fullerene acceptors (NFAs)[3], e.g., Y6[4] and its derivatives, further advanced the limits of the power conversion efficiency of OSCs, with recorded efficiencies recently exceeding 20% for the first time[4]. Recent progress in overcoming the stability issues associated with heterojunctions[5] has also been made via the synthesis and characterisation of single-component OSCs[6,7].

Theory and computation are lagging behind those recent experimental developments leaving a large gap in our fundamental understanding of charge generation in OSCs[8]. Whilst the initial formation of bound electron-hole pairs or Frenkel excitons upon light absorption in low dielectric organic semiconductor (OS) materials is generally agreed upon, their subsequent conversion to free charge carriers is less clear. Two principal mechanisms have been postulated in the literature: the so-called cold exciton dissociation mechanism, where an exciton dissociates at the donor-acceptor interface into a low-energy (cold) charge transfer (CT) state (in the following denoted interfacial CT (iCT) state) that needs to overcome the large Coulomb barrier via thermal motion to form free charges[9–12]. In the second mechanism, termed hot exciton dissociation[2,13–18], the exciton instead dissociates into electronically excited (hot), non-interfacial CT (niCT) states, which are poised to transition into charge separated (CS) states and free charge carriers.

The use of density functional theory (DFT) has been instrumental in describing opto-electronic properties of organic molecules and donor-acceptor complexes[19–22]. Moreover, assuming that the dynamics

[1]Department of Physics and Astronomy and Thomas Young Centre, University College London, London, UK. [2]Department of Chemistry and Industrial Chemistry, University of Pisa, Pisa, Italy. [3]Department of Chemistry, Tunghai University, Taichung City, Xitun District, Taiwan, ROC. ✉e-mail: j.blumberger@ucl.ac.uk

of photo-excited OSs can be treated as a sequence of transitions between localised molecular electronic states with well-defined rate constants (e.g. Marcus or Redfield rates), Kinetic Monte Carlo (KMC) simulations have found some success in reproducing internal quantum efficiencies (IQEs) of certain systems[23–25]. However, ordered OSs, especially the ones that would permit large IQEs, are in a difficult regime where the assumption of discrete transitions between localised molecular electronic states may no longer be a valid approximation. Indeed, recent work has shown that electronic state delocalisation[26–37], the symmetry of electronic band states (or, equivalently, the sign-pattern of electronic couplings)[38,39], and the feedback from the electronic to the nuclear subsystem[40–42] are essential for a realistic description of charge carrier[26–29,38,39] and exciton transport dynamics[30,31,35,40,42] in ordered OSs - these are quantum effects that are missing in typical KMC simulations. This is not expected to be any different for charge generation in the latest generation of high efficiency OSC materials, as they tend to form highly crystalline morphologies[4]. Hence, the development of simulation methods that include these vital effects is highly desirable.

In this respect, a study by Scholes considered the energetics of delocalised eigenstates of a static site-based Hamiltonian, and concluded that the energetic alignment of the exciton and niCT states promotes free charge generation, and avoids the occupation of the kinetically trapped iCT state[43]. Further support for this finding requires the prediction of dynamics, specifically the propagation of time-dependent populations of the electronic states involved in charge generation, which has since been attempted with a variety of methods with varying degrees of accuracy. Gélinas et al. modelled their experimental evidence of ultrafast charge separation in a fullerene acceptor heterojunction by propagating the electronic state populations of a similar site-based Hamiltonian with a truncated state space, and concluded that charge generation in such a blend is assisted by the transient occupation of high-energy niCT states with a delocalised anion[14]. A notable development was made by Vukmirović and Janković[44], who designed an approach to describe the time-dependent population dynamics of delocalised electronic states, starting from a Hamiltonian constructed from the full manifold of localised singlet exciton/CT-states in a model heterojunction. A further significant advancement, based on a similar formulation of the electronic structure, was recently achieved by Kassal and coworkers, who simulated the population dynamics of delocalised quantum states with an analogue to KMC termed delocalised Kinetic Monte Carlo (dKMC)[32–34].

Non-adiabatic dynamics approaches are a powerful alternative to the above methods and are currently among the most accurate approaches for the simulation of the electronic quantum dynamics of photo-excited organic materials. Multi-configurational Time-dependent Hartree (MCTDH)[45–47] is generally viewed as the gold standard, but this method is computationally expensive even in its multilayer version. Here we report on the development and application of eXcitonic state-based surface hopping (X-SH), a computationally less expensive quantum-classical non-adiabatic dynamics method enabling realistic atomistic simulation of photo-induced charge generation in truly nanoscale donor-acceptor interfaces (10 − 100 nm) on experimentally relevant time scales ( > 10 ps). Solving the electronic Schrödinger equation coupled to the finite temperature nuclear motion, the method naturally accounts for delocalisation of excitons and charge carriers, diagonal and off-diagonal thermal electronic disorder (as well as static disorder if present) and the feedback from electronic to nuclear dynamics. Our method comprises a major improvement over a preliminary version of X-SH that did not account for off-diagonal thermal electronic disorder[48] deemed essential for exciton[30,31,35,40,42] and charge transport[26–29,38,39] and thus for photo-initiated charge generation.

We apply our simulation tool to systematically investigate the effect of important material parameters on the mechanism and efficiency of exciton dissociation—in particular, the electronic coupling between the molecules in donor and acceptor materials (determining their band width and giving rise to charge carrier delocalisation), and the electron-hole interaction strength (related to their dielectric constant and exciton binding energy). To this end we consider a model heterojunction comprised of a crystalline donor, α-hexathiophene (a6T), and a crystalline NFA, perylene-diimide (PDI) whose derivatives have been previously characterised as components of heterojunction OSCs[49,50]. Our choice to consider an idealised crystalline system reflects the fact that the latest organic photovoltaic devices with record efficiencies are made of crystalline materials[4]. Electronic coupling and electron-hole interaction strength are initially chosen to correspond to the physical values as obtained by DFT parametrisation. These parameters are then varied within physically realistic bounds for OSs to explore different charge generation regimes while keeping the atomistic structure of the molecules and the packing structure of the interface unchanged. In this way, the effect each parameter has on exciton dissociation can be identified in isolation. We lastly examine the role played by structural disorder by selecting parameters that led to efficient charge generation in the crystalline a6T:PDI interface, and re-simulating the exciton dissociation process with an interfacial defect introduced to disrupt the donor/acceptor stacking pattern.

We find that for the a6T:PDI interface, charge generation proceeds via the cold exciton dissociation mechanism: fast relaxation of the initial band-like electronic excitation to a localised Frenkel exciton, diffusion of the Frenkel exciton to the interface via hopping transport, followed by formation of a vibronically fully relaxed iCT state that slowly dissociates thermally to free carriers. Intriguingly, as we increase the electronic coupling between the molecules we increasingly observe a second, more efficient hot exciton dissociation channel. Here, Frenkel excitons convert to transiently delocalised hybrid exciton-charge transfer states (hybrid XT-CT states) that directly transition into niCT and CS states. Remarkably, the hot exciton dissociation process is observed to occur at distances of up to several nanometers away from the interface owing to the delocalised nature of hybrid XT-CT states. This way, the formation of trapped iCT states that are prone to non-radiative recombination is avoided. Similar observations are made when the dielectric constant of the donor and acceptor is increased in place of electronic coupling. Additionally, for a parameter combination where the hot pathway dominates, the insertion of an interfacial defect into the a6T:PDI heterojunction significantly reduces the rate of exciton dissociation.

## Results and discussion

### Model electronic Hamiltonian for a6T:PDI heterojunction

Quantum chemical methods (e.g., TDDFT) generate electronically excited adiabatic states but they are too time-consuming for non-adiabatic dynamics of large molecular systems. A more feasible strategy is to fragmentise the system into molecules or molecular fragments and to construct the electronic Hamiltonian of the full system in the space of locally excited or quasi-diabatic electronic states, which we refer to as local states in the following. This is illustrated here for the a6T:PDI heterojunction that is schematically shown in Fig. 1a–c. It contains a 1-dimensional electronically active region of 20 a6T donor and 20 PDI acceptor molecules embedded within an extended 3D crystalline environment that ensures the structural integrity of the electronically active system (see Methods for details of structure generation). We consider a minimal electronic state space for this system consisting of (i) the lowest singlet excited states $|\phi_k^{XT}\rangle$ where the excitation is localised on molecule $k$ while all other molecules are in their ground state, (ii) the charge transfer (CT) states $|\phi_{kl}^{CT}\rangle$, where molecule $k$ carries a positive charge (hole) and molecule $l$ a negative charge (excess electron) while all other molecules are in their ground state. The corresponding electronic Hamiltonian in this locally excited

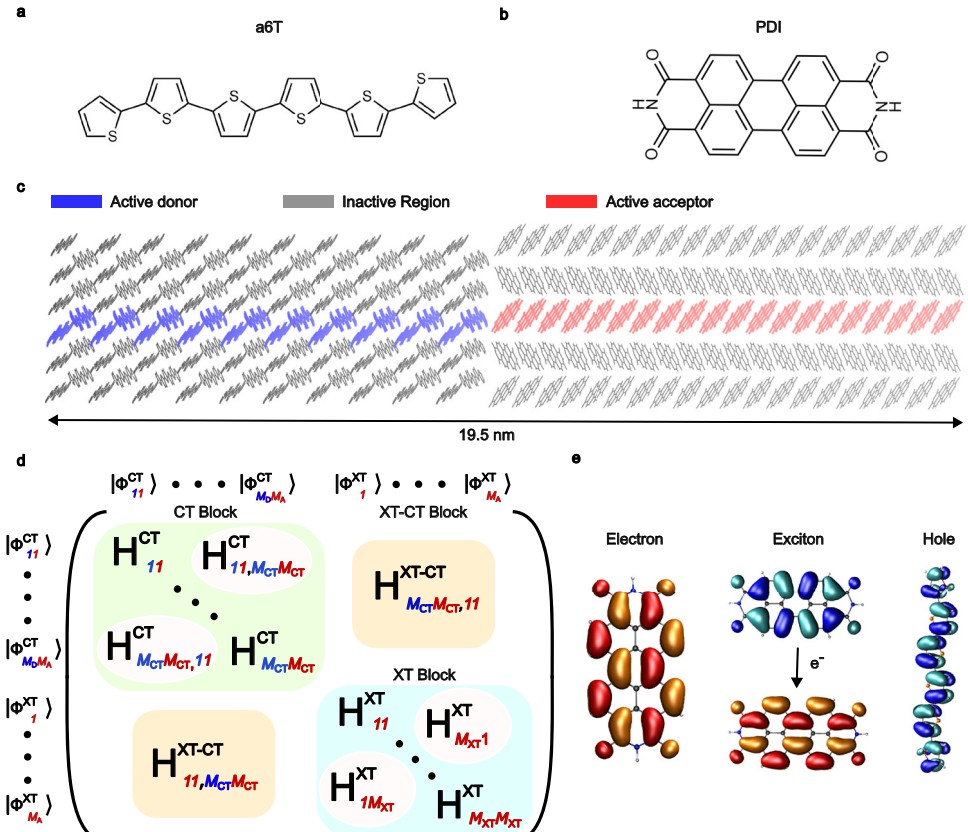

**Fig. 1 | Pictorial representation of the atomistic a6T:PDI heterojunction interface and electronic structure. a**, **b** show the skeletal formulae of the a6T and PDI molecules, respectively. **c** displays the full heterojunction interface, with the coloured regions indicating where the charges and excitons are explicitly propagated, whilst the grey regions exchange energy with the former via Lennard-Jones interactions. A schematic of the electronic Hamiltonian is shown in **d**, with two diagonal blocks being assigned to localised CT and XT states' energies and couplings; the off-diagonal block corresponds to the coupling between CT and XT states. The (quasi-)diabatic wavefunctions, given as $\phi^{CT/XT}$, of the localised XT/CT states corresponding to each matrix element are illustrated above and to the left of the matrix. The wavefunction indices denote donor/acceptor molecules, whilst the indices of the matrix elements denote the rows and columns of the XT/CT blocks in terms of the total number of XT/CT states. **e** displays the frontier orbitals that contain the localised charges or excitons.

state basis is given by Eq. (2) and shown schematically in Fig. 1d. The diagonal elements of $\widehat{H}^{XT}$ and $\widehat{H}^{CT}$ are referred to as site energies (Eqs. (9) and (11)) whilst the off-diagonal elements are the excitonic couplings between XT states (Eq. (12)) and the electronic couplings between the CT states, respectively (Eq. (16)), and $\widehat{H}^{XT-CT}$ contains the couplings between XT and CT states. While in our present application we only consider singlet excited states, it is relatively straightforward to extend the electronic state space of X-SH to higher spin and higher energetic states if required.

We further assert that charge generation is dominated by excitons formed in the acceptor phase. This is hinted by our previous finding that exciton transport is faster in PDI than in a6T[40]. Hence, we disregard exciton formation in the a6T phase. The excitonic state of PDI molecule $i$, $|\phi_i^{XT}\rangle$, is described by the lowest S1 singlet excitation which is dominated by the natural transition orbitals that closely resemble the highest occupied molecular orbital (HOMO) and the lowest unoccupied molecular orbital (LUMO) (98.6%, see Fig. 1c in ref. 40). Excitations above the first-excited singlet state are relatively high in energy and are disregarded. The frontier orbitals relevant to Frenkel exciton formation in PDI, and for electron and hole transport in PDI (LUMO) and a6T (HOMO), respectively, are shown in Fig. 1e. In regard to the CT states, TDDFT calculations predict that the interfacial CT state where the excess electron and hole are localised on the interfacial a6T:PDI pair is 390 meV below the Frenkel exciton state that is localised on the interfacial PDI molecule (see Methods). The electronic energy of the CT states further away from the interface is based on the interfacial CT state energy plus a screened Coulomb electron-hole interaction term (dielectric constant $\epsilon_r = 3.5$, a typical value in organic semiconductors[51]) and an external electric field contribution typically found in OSCs ($10^5$ V cm$^{-1}$)[20,52]. This results in an exciton binding energy (defined in Eq. (24)) of 260 meV, which is typical of NFA materials[53]. Excitonic and electronic couplings between the respective states are calculated efficiently using the transition charge (TrESP[54]) and analytic overlap methods (AOM[55,56]) parameterised to (TD)DFT calculations, respectively. This enables the efficient update of all electronic Hamiltonian matrix elements (site energies and couplings) at each nuclear time step of the X-SH simulations, thereby accounting for diagonal and off-diagonal thermal electronic disorder (related to the electron-phonon coupling). Relevant parameters defining the electronic Hamiltonian and the diagonal and off-diagonal thermal electronic disorder are given in Table 1, with further details given in the Methods section.

## Density of electronic states

We first examine the electronic structure of the a6T:PDI junction at 0 K by diagonalising the electronic Hamiltonian at the minimum energy configuration of the junction. This generates the adiabatic electronic eigenstates of the system as a linear superposition of the local XT and CT states, as shown in Fig. 2a (0 K distributions in full colours). To classify the eigenstates we first divide the local CT states into iCT, niCT and CS states. In the iCT state the electron-hole pair resides solely at the donor and acceptor molecules forming the interface, in the niCT

state between the interface and the maximum of the Coulomb barrier (that results from the applied external field) and in the CS state beyond the Coulomb barrier maximum. The eigenstates are then denoted XT (CT) eigenstates if they project to > 95% on local XT (CT) states and as mixed hybrid XT-CT states otherwise. CT eigenstates are then further divided into iCT (CS) eigenstates if they project to > 80% on local iCT (CS) states and as niCT eigenstates otherwise.

**Table 1 | Key parameters governing exciton transfer (XT), electron transfer (ET), hole transfer (HT) and exciton dissociation to charge transfer states (XT-CT) in the a6T:PDI interface[a]**

| transfer type $S$ | phase | $\lambda_i^{S\ b}$ (meV) | $\langle |H_{kl}^S| \rangle^c$ (meV) | $\sigma_{kl}^{S,d}$ (meV) |
|---|---|---|---|---|
| ET | PDI | 255.8 | 69.2 | 61.6 |
| HT | a6T | 253.7 | 43.5 | 12.0 |
| XT | PDI | 390.5 | 87.8 | 5.88 |
| XT-CT | PDI, a6T | 186.4 | 59.6 | 30.7 |

[a]Other electronic parameters: dielectric constant $\epsilon_r = 3.5$ determining the Coulomb barrier, Eq. (11), energy offset between interfacial XT and interfacial CT diabatic states $\epsilon^{XT} = 390$ meV (Eq. (9)), exciton binding energy $E_b = 260$ meV (Eq. (24)).

[b]Reorganisation energies, Eq. (10).

[c]Electronic or excitonic couplings obtained by averaging over 500 10ps-long X-SH trajectories. For XT-CT coupling, the average excited state electronic coupling, $\langle |H_{k,mn}^{XT-CT}| \rangle$, is reported.

[d]Thermal fluctuations of electronic or excitonic couplings, $\sigma_{kl}^S = \left( \left\langle \left( H_{kl}^S - \langle H_{kl}^S \rangle \right)^2 \right\rangle \right)^{1/2}$. For XT-CT coupling, the thermal fluctuation of the excited state electronic coupling,

$\sigma_{k,mn}^{XT-CT} = \left( \left\langle \left( H_{k,mn}^{XT-CT} - \langle H_{k,mn}^{XT-CT} \rangle \right)^2 \right\rangle \right)^{1/2}$ is reported.

We find that the centre of the XT band (black) remains at ≈ 400 meV above the lowest energy iCT states (red) with a band width that is $4H_{kl}^{XT}$, where $H_{kl}^{XT}$ is the excitonic coupling between PDI molecules in the direction orthogonal to the interface. The niCT and CS bands are at higher energies than the XT band, about 650 meV above iCT, with a band width greater than $4[(H_{km}^{HT} + H_{ln}^{ET})/2]$, where $H_{km}^{HT}$ and $H_{ln}^{ET}$ are the electronic couplings for hole and electron transfer in a6T and PDI in the direction orthogonal to the interface, respectively. Evidently, the Coulomb interaction (shown in orange in Fig. 2) results in CT bands that are wider than what one would expect from electronic coupling interactions alone. Notice that the centre of the CS band is below that of the niCT band due to the applied electric field (Fig. 2a solid distributions). Notice also that the density of hybrid XT-CT states is small due to the energetic separation of XT and CT bands resulting in little mixing. MD simulations in the electronic ground state show that all bands become considerably broadened at 300 K (Fig. 2a shaded distributions).

### Cold exciton dissociation

The time-dependent relaxation of a photo-induced excitation in the electronic state space of the a6T:PDI junction characterised above is simulated using X-SH. We refer to Methods for a description of the implemented X-SH method and for simulation details. Photo-excitation is modelled by a Gaussian laser pulse generating an initial distribution of adiabatic eigenstates within the XT band that are propagated for 10 picoseconds each. The time evolution of the electronic wavefunction $\Psi(t)$ (Eq. (6)) along a representative X-SH trajectory is shown in Fig. 3. After initial excitation into a delocalised excitonic eigenstate in the PDI phase (Fig. 3a, inverse participation ratio (IPR, Eq. (21)) ≈ 5), the relaxation process can be broadly resolved into 4 stages.

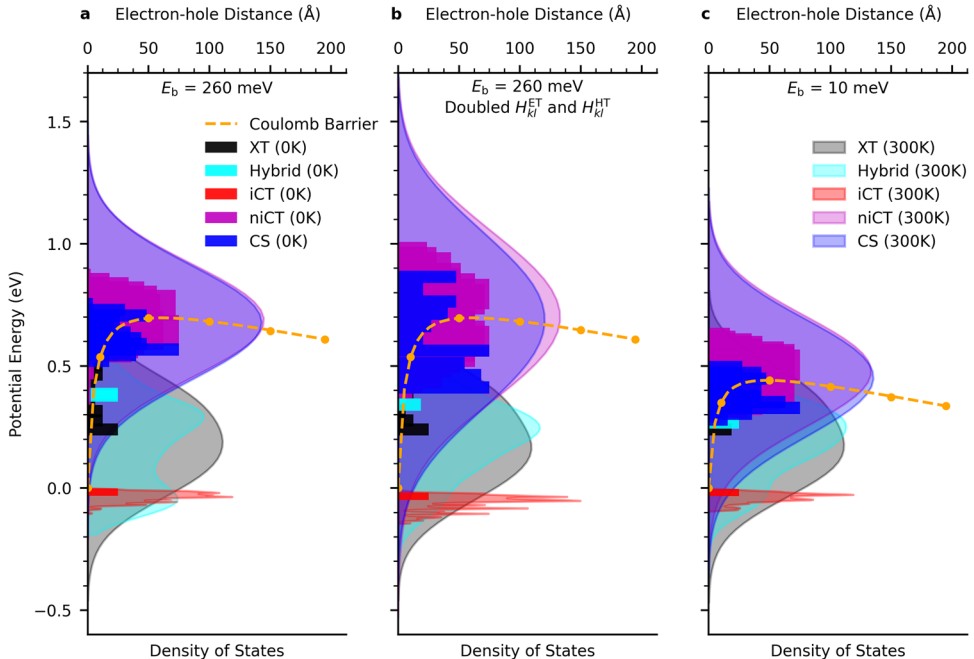

**Fig. 2 | Densities of different types of eigenstates of the a6T:PDI heterojunction.** Densities of eigenstates resulting from a thermal distribution of a6T-PDI geometries in the neutral ground state are shown by the shaded areas, whilst that of the 0 K structure is shown in the full-coloured blocks. The figure displays the densities of states for the junction with unmodified couplings and an exciton binding energy of 260 meV (**a**), a doubled magnitude of electronic coupling between localised CT states (**b**), and a reduced binding energy of 10 meV (**c**). The relevant potential energy barrier due to the Coulombic interaction of the electron-hole pair, and its interaction with an external potential bias, is displayed in orange on each panel. The electron-hole distance (in Å), relative to that between the interfacial electron-hole pair, is included as an x-axis on the top of each panel, whilst the density of states is on the bottom. The exciton binding energy is adjusted via the tuning of the strength of the Coulombic interaction. The densities of eigenstates obtained from thermal distributions were re-normalised for each band, due to the large difference between the number of CT and XT states in the X-SH state space. Increased energetic overlap between the lower XT band edge and the niCT/CS bands can be achieved by doubling the electronic coupling between CT-states (**a** vs **b**) or by increasing the dielectric constant from 3.5 to 5 (**a** vs **c**).

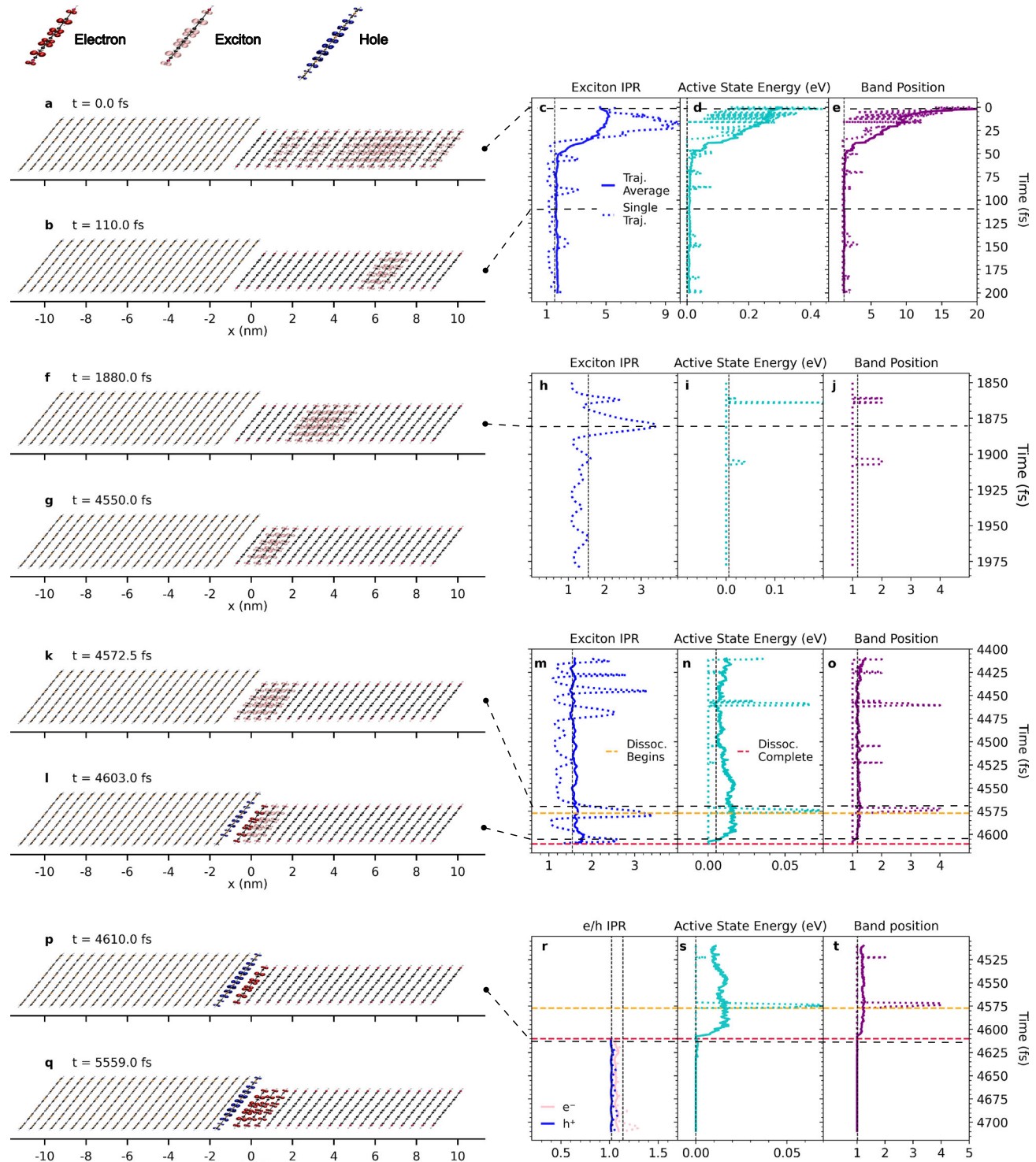

**Fig. 3 | Evolution of the wavefunction of an excitation in the cold exciton dissociation regime.** Left-hand side panels pictorially represent the evolution of an excitation in a typical trajectory of X-SH, when the a6T:PDI junction has $E_b$ = 260 meV and unmodified electronic couplings. The extent of charge/exciton localisation on each molecule was visualised via the plotting of charge/transition density isosurfaces, whose amplitudes were scaled by the relative probabilities of these molecular sites being occupied. The evolution can be divided into 4 stages: 1. Initial excitation of exciton, followed by relaxation and localisation (**a**, **b**). 2. Exciton diffusion to the heterojunction interface assisted by transient delocalisation (**f**, **g**).

3. Transient delocalisation of the exciton and hybrid XT-CT state formation (**k**, **l**). 4. Full exciton dissociation into an interfacial CT state, which fails to surmount the strong Coulombic electron-hole interaction (**p**, **q**). Panels on the right-hand side plot the evolution of relevant observables over the course of the key events shown on the left. This includes: exciton IPR (**c**, **h**, **m**); active eigenstate potential energy with respect to the lowest-energy state in the relevant band (**d**, **i**, **n**, **s**); active eigenstate band position with respect to the band's lowest-energy state (**e**, **j**, **o**, **t**); electron and hole IPR (**r**). The vertical dashed lines correspond to observable averages over all timesteps and all trajectories.

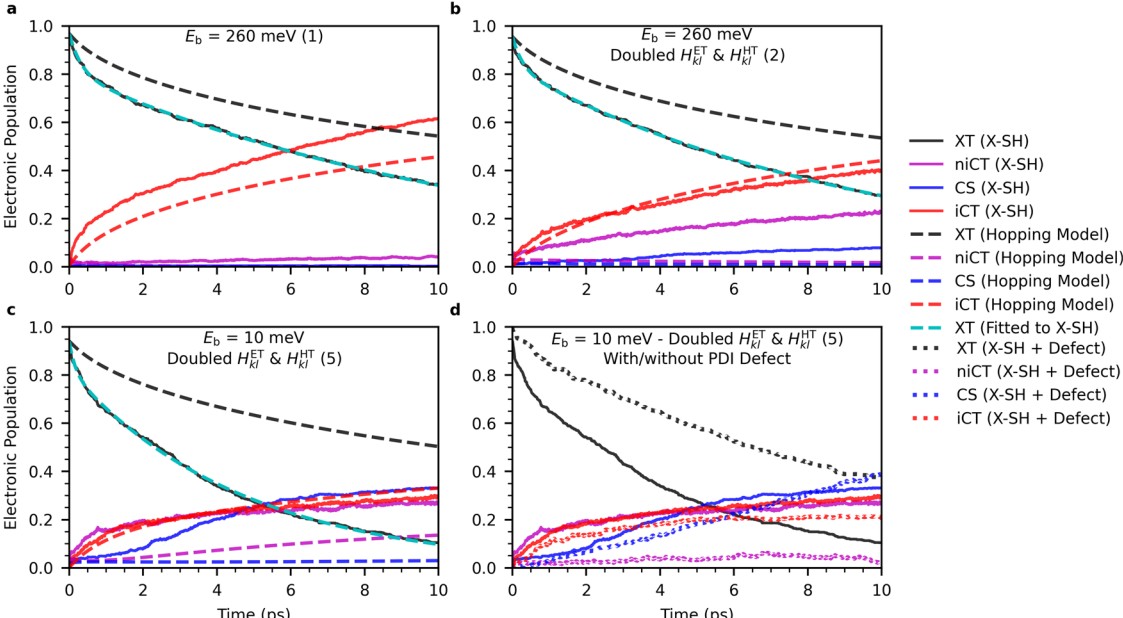

**Fig. 4 | Kinetics of exciton dissociation and charge generation. a–c**: Populations for XT (Eq. (17)), iCT (Eq. (18)), niCT (Eq. (19)) and CS (Eq. (20)) states are plotted as a function of time as obtained from X-SH simulation of a fully ordered heterojunction (solid lines) or by solving a chemical Master equation (dashed lines) for discrete transitions between the diabatic electronic states. **a** is for the unmodified interface (Simulation 1 in Table 2), **b** for doubled electronic coupling (Simulation 2 in Table 2) and **c** for doubled electronic coupling and a dielectric constant increased to 5 resulting in a decrease of exciton binding energy to 10 meV (Simulation 5 in Table 2). Fits of the XT population to the bi-exponential function Eq. (1) is shown in dashed cyan lines. Notice the good agreement between X-SH and the chemical Master equation approach for CT states in **a**, but the large discrepancy in niCT and CS populations in regimes shown in **b**, **c**, as the Master equation approach does not account for the formation of hybrid XT-CT states and charge delocalisation. **d** compares the population dynamics of the fully ordered heterojunction (solid lines, electronic coupling doubled and dielectric constant set to 5) with a similar system containing an interfacial defect (dotted lines).

First, rapid exciton relaxation to the bottom of the XT band leading to exciton localisation on one to two molecules (IPR = 1–2) within 50 fs (Fig. 3b and c). The dissipated excitation energy of about 290 meV is due to fast downward electronic transitions within the XT band (Fig. 3d and e) and the geometric relaxation related to an internal reorganisation energy of a single PDI molecule in the S1 state of 195 meV.

Second, diffusion of the exciton to the a6T:PDI interface on the ps time scale assisted by a transient delocalisation mechanism (Fig. 3f and g). Here, wavefunction contraction and expansion events (Fig. 3h) induced by short-lived thermal excitations within the excitonic band (Fig. 3i and j) eventually bring the exciton into the interfacial region.

Third, exciton dissociation to the iCT state. When thermal fluctuations bring the interfacial excitonic wavefunction (Fig. 3k) in close energetic alignment with the iCT state, the two states start to mix due to the finite electronic coupling between local iCT and local interfacial XT states. This results in the formation of a localised, low-energy hybrid XT-CT state at the interface (Fig. 3l–o) that rapidly converts to the iCT state (Fig. 3p) within 20–30 fs. The iCT state remains solely at the bottom of the CT band due to the strong Coulomb interaction, which localises the charges and allows the interfacial a6T:PDI molecules to relax to their equilibrium geometries in the cationic/anionic charge states.

Finally, charge separation or recombination of the iCT state. We observe frequent oscillations of charge density between iCT and niCT states (Fig. 3q) but the electron-hole pair is ultimately unable to overcome the large Coulomb barrier (700 meV) between the vibronically fully relaxed iCT state and the CS states on the 10 ps time scale of present X-SH simulations and remains at the bottom of the CT band (Fig. 3r–t). The subsequent dynamics, that is charge separation or recombination of the iCT state, involves longer time scales. This dynamics is not included in the present simulations as these longer time scales cannot be presently accessed. The mechanism described above is referred to as cold exciton dissociation because all excitons relax to the lowest excited (cold) state of the system, i.e. iCT state, before they acquire enough thermal energy to separate to free charges or recombine to the ground state.

To describe the relaxation dynamics more quantitatively, we project $\Psi(t)$ onto the local XT, iCT, niCT and CS states and average over all trajectories to obtain the corresponding time-dependent populations $P_{XT}$, $P_{iCT}$, $P_{niCT}$ and $P_{CS}$, (given by Eqs. (17–20)). We find that the decay for XT fits well a bi-exponential model (Fig. 4a, solid black and dashed cyan lines, $R^2 = 0.9991$),

$$P_{XT} = C_i \exp(-k_i t) + C_b \exp(-k_b t), \qquad (1)$$

where $k_i$ and $k_b$ can be interpreted as rate constants for the dissociation of excitons that are photo-generated in the vicinity of the interface (subscript i) and the bulk region of PDI (subscript b), respectively. The interface-generated excitons can readily dissociate to iCT states without having to diffuse across the PDI phase giving rise to a fast rate constant $k_i = 3.03 \text{ ps}^{-1}$. The bulk-generated excitons decay about a factor of 30 slower, $k_b = 8.65 \times 10^{-2} \text{ ps}^{-1}$, indicating that diffusion to the interface is rate-limiting.

The contribution of interface and bulk-generated excitons in our simulation, $C_i = 0.15$ and $C_b = 0.82$, reflects the 20 times larger number of PDI molecules in the bulk phase compared to the interface. In general, the contributions will depend on the interface surface to volume ratio of the heterojunction. We note that the resolution of the exciton decay profile into interfacial and diffusion-limited components has also been observed in experiment[14,57]. The decay of the XT population is mirrored by an increase in the CT population where, in accord with the above example trajectory, the vast majority of the CT population on the present simulation time scale is composed of iCT states (Fig. 4a, solid red lines). Only traces of niCT (solid purple) and virtually no CS states (solid blue) are observed.

**Table 2 | Exciton decay constants and charge separation yields of the model a6T:PDI heterojunction in different parameter regimes from X-SH simulations[a]**

| Simulation | $\epsilon_r^b$ | $E_b^c$ (meV) | $\langle |H_{kl}^{HT}| \rangle^d$ (meV) | $\langle |H_{kl}^{ET}| \rangle^d$ (meV) | $k_i^e$ (ps$^{-1}$) | $k_b^e$ (ps$^{-1}$) | CS yield$^f$ (%) |
|---|---|---|---|---|---|---|---|
| (1) | 3.5 | 260 | 43.5 | 69.2 | 3.03 | 0.086 | 0.28 |
| (2) | 3.5 | 260 | 87.2 | 150.1 | 3.53 | 0.103 | 7.96 |
| (3) | 3.5 | 260 | 131.8 | 251.3 | 5.69 | 0.179 | 25.2 |
| (4) | 5 | 10 | 43.5 | 69.9 | 3.03 | 0.093 | 2.61 |
| (5) | 5 | 10 | 87.4 | 160.8 | 6.08 | 0.213 | 33.1 |
| (6) | 5 | 10 | 132.2 | 260.9 | 9.31 | 0.244 | 44.94 |
| (7) | 10 | -310 | 43.5 | 73.6 | 7.21 | 0.297 | 17.1 |

[a]Model parameters as in Table 1 unless specified otherwise. (1) is the physical (unmodified) a6T:PDI junction. In (2) and (3) the constants of proportion between orbital overlap and electronic coupling for HT and ET, $\bar{C}^{HT}$ and $\bar{C}^{ET}$, are scaled by a factor of 2 and 3, respectively, resulting in thermal averages for $H_{kl}^{ET}$ and $H_{kl}^{HT}$ that are about a factor of 2 and 3 higher than in simulation (1). See section Methods for a definition of $\bar{C}^{HT}$ and $\bar{C}^{ET}$. In (4), (5) and (6) the dielectric constant $\epsilon_r$ is increased to 5 for unmodified and scaled electronic couplings and in (7) to a value of 10 for unmodified electronic couplings.
[b]Dielectric constant determining the Coulomb barrier, Eq. (11).
[c]Exciton binding energy, Eq. (24).
[d]Electronic couplings obtained by averaging over 500 10 ps-long X-SH trajectories.
[e]Exciton decay constants for the decay of interface-generated ($k_i$) and bulk-generated excitons ($k_b$), obtained by fitting the XT state population $P_{XT}$ (Eq. (17)) from X-SH to Eq. (1).
[f]Charge separation (CS) yield defined as the percentage of 500 X-SH trajectories whose electronic wavefunction $\Psi(t)$ is classified as a CS state after 10 ps of X-SH dynamics.

## Hot exciton dissociation

The above results imply that in order to accelerate free charge generation, the formation of iCT trap states should be avoided and excitons should preferentially relax to niCT and on to CS states. In the a6T:PDI heterojunction studied above, this does not happen because the niCT and CS bands are too high in energy compared to the XT band (see Fig. 2a). In the following we consider two strategies to achieve a better energetic overlap between XT and niCT/CS band states: (i) increase of the electronic coupling in the donor and acceptor materials and (ii) increase of the dielectric constant. The newly assigned parameters defining the electronic Hamiltonian after taking approaches (i) and (ii) are shown in Table 2. All other parameters remain the same as in the simulations of the original system above. We find that doubling the electronic couplings (Table 2, simulation 2) and thus the width of niCT and CS bands creates tail states that are close in energy with low-energy states in the XT band (Fig. 2b). Alternatively, a small increase in dielectric constant from 3.5 to 5 (Table 2, simulation 4) shifts the niCT and CS bands down in energy to align with the XT band (Fig. 2c). Adopting the common expression for exciton binding energy[53] (Eq. (24)) and approximating the formation energy of free charges from the iCT state by a screened Coulomb potential, we find that the increase in dielectric constant corresponds to a lowering of the exciton binding energy from 260 to 10 meV (however, we note that this is an approximation as the electron-hole interaction energy at short range may not exactly follow a screened Coulomb interaction). In both cases, with an increase of the electronic coupling or dielectric constant, the resulting close energetic alignment also increases the density of hybrid XT-CT states.

We have carried out X-SH simulations for all parameter regimes denoted simulations 2–7 in Table 2. The population dynamics corresponding to a doubled electronic coupling magnitude (Table 2, simulation 2) is shown in Fig. 4b. We find that the exciton decay remains bi-exponential ($R^2 = 0.9991$) with $k_i$ and $k_b$ now marginally larger than in the previous simulations with unmodified electronic couplings. Importantly, the CS state population is now 8.0% at 10 ps, up from 0.3% for unmodified couplings. If both the electronic couplings are doubled and the dielectric constant is increased from 3.5

to 5 (Table 2, simulation 5), the exciton dissociation rate increases further, generating the CS state as the majority species after 10 ps, 33.1%, Fig. 4c. The CS yield is increased further if electronic couplings are tripled and/or the dielectric constant is increased to 10, see Supplementary Fig. 2. However, this high coupling/high dielectric regime is not easily accessed with conventional organic semiconductors.

The cause of the remarkable increase in CS yield is now a pertinent question. Analysing the X-SH trajectories, we find that the improved energetic alignment of XT and CT states brought about by larger electronic couplings and/or higher dielectric constant gives rise to an additional and more efficient hot exciton dissociation mechanism. This is exemplified by the evolution of the wavefunction $\Psi(t)$ along a representative X-SH trajectory for doubled electronic coupling and dielectric constant increased to 5 (simulation 5, Fig. 5). Whilst the initial stages of exciton relaxation are the same as in the cold exciton dissociation mechanism (Fig. 5a), we now observe the formation of a state not seen before, whereby a transiently delocalised XT state (Fig. 5b) forms a delocalised hybrid XT-CT state (Fig. 5c). This state is a superposition of delocalised XT, iCT and niCT states. In such states, the excitons and charges are higher in energy and delocalised to a greater extent (see Fig. 5d–f and Supplementary Fig. 6) and can be centred further from the donor/acceptor interface (Supplementary Fig. 8). It is short-lived and rapidly converts into a niCT state that exhibits significant amplitudes on a6T and PDI molecules that are 2–3 lattice spacings away from the interface (Fig. 5g), which can then transition to a CS state (Fig. 5h). The larger mean separation distance significantly reduces the Coulomb electron-hole attraction compared with the iCT state (see Coulomb barrier indicated in orange in Fig. 2c). The increased occupancy of niCT states is reflected in the higher energy and band position of the active eigenstate in X-SH after exciton dissociation is completed (Fig. 5j and k) relative to the iCT state formed in the cold dissociation pathway (Fig. 3s and t). In addition to this, the electron and hole in the niCT state are delocalised over 2–3 molecules as a consequence of the doubled electronic couplings (Fig. 5i) which impedes the full intramolecular relaxation of the partially charged a6T and PDI molecules. As the remaining Coulomb attraction can be relatively easily overcome through thermal fluctuations, the niCT state is poised for charge separation.

A remarkable feature of the hot exciton dissociation mechanism is that a significant proportion of excitons no longer dissociate at the interface, as traditional models of excitonic solar cells would assume[2], but several lattice spacings/nanometres away from it, owing to the delocalised nature of hybrid XT-CT states in this regime (see Supplementary Figs. 4 and 10, using the center of the wavefunction as a distance measure). Notice that the iCT states are still (on average) the energetically lowest excited states and they are still formed (Fig. 4c). Thus, in this regime both the cold and the hot dissociation mechanisms occur simultaneously (Supplementary Fig. 5). Yet, the fraction of excitons dissociating via the efficient hot mechanism steadily increases with increasing electronic coupling and/or dielectric constant.

## Hopping Model vs eXcitonic state-based Surface Hopping

Charge generation in OSCs is often modelled with KMC or, equivalently, chemical master equations using excitation energy transfer and electron transfer rates for exciton or charge hopping between different molecules. In the following we investigate how the population dynamics obtained from a chemical Master Equation compares with the ones obtained from X-SH simulations. Hopping rates between sites were calculated with semiclassical Marcus theory[58], with the relevant electronic parameters (reorganisation energies, electronic or excitonic coupling and driving force) chosen to be the same as in X-SH simulations. Full details of the calculations are given in Supplementary Discussion 1. The results for the original system where only the cold

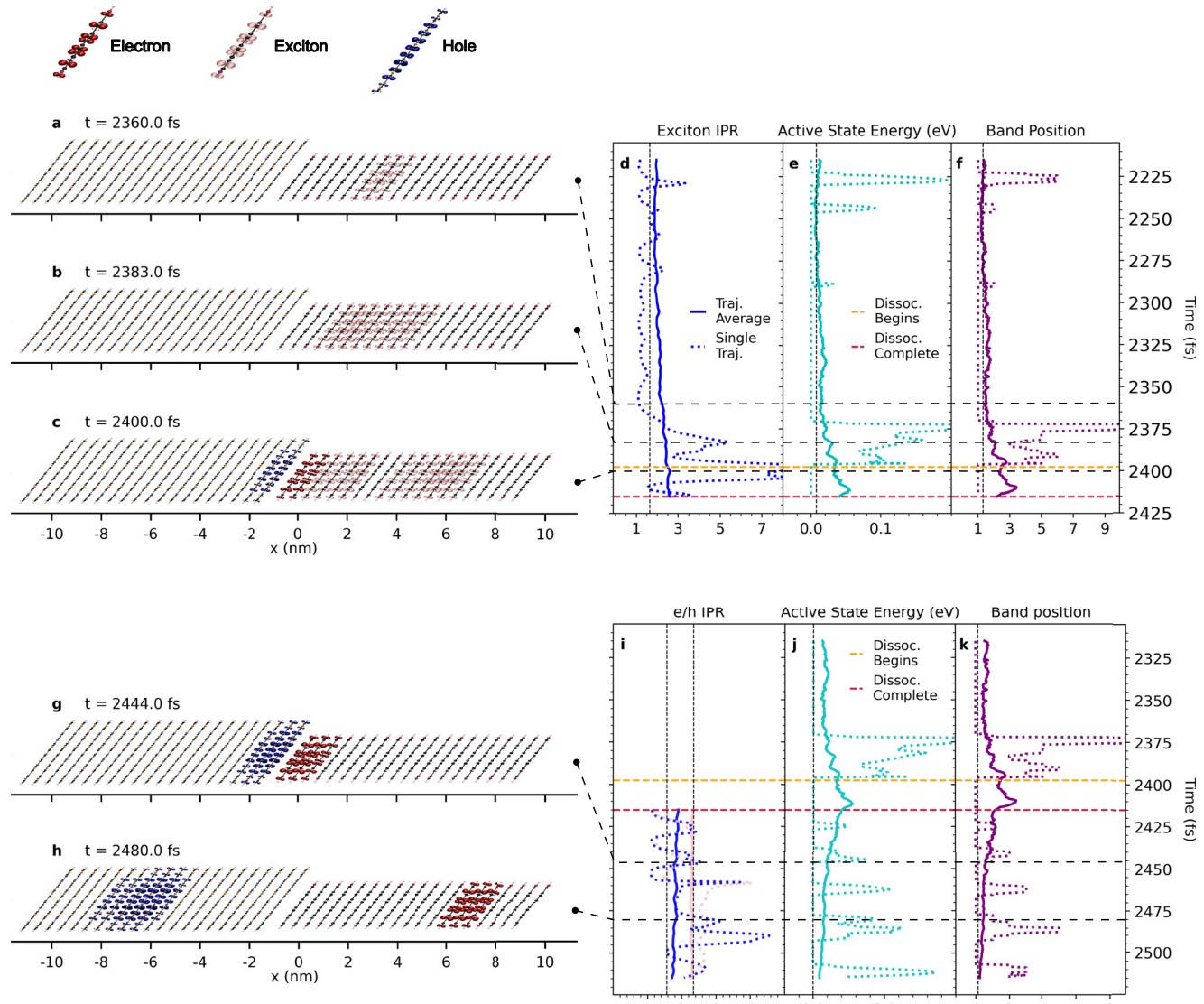

**Fig. 5 | Evolution of the wavefunction of an excitation in the hot exciton dissociation regime.** The panels on the left-hand side of the figure demonstrate the evolution of an excitation in a common X-SH trajectory, where $E_b$ has been reduced to 10 meV and the electronic coupling between CT states has been doubled (simulation 5 in Table 2). A non-interfacial, localised exciton (**a**) is now able to transiently delocalise (**b**) and transition to a delocalised hybrid state with non-interfacial XT and CT character (**c**). This transiently delocalised hybrid XT-CT state then converts into an niCT state (**g**), which is able to transition to an energetically accessible CS state (**h**). As in Fig. 3, the right-hand side panels plot the evolution of relevant observables over the time frames of the events shown on the left. This includes: exciton IPR (**d**); active eigenstate potential energy relative to the lowest-energy state in the relevant band (**e, j**); active eigenstate position relative to the lowest-energy state in the band (**f, k**); electron and hole IPR (**i**). The vertical dashed lines correspond to observable averages over all timesteps and all trajectories.

exciton dissociation mechanism operates (simulation 1 in Table 2) is shown in Fig. 4a (dashed lines). Although the exciton dissociation rate is somewhat underestimated compared to X-SH, the master equation approach faithfully models the population dynamics of CT states. The reasonably good agreement is not unexpected because in this regime (strong Coulomb interaction, moderate couplings) the excitons dissociate to localised charge carriers at the interface, as assumed in the master equation approach. However, when electronic couplings and/or dielectric constant are increased the populations of niCT and CS states increase only slightly (simulations 2 and 5 in Table 2, dashed lines in Fig. 4b and c), severely underestimating the large increase obtained from X-SH. In fact, the population dynamics appears to have near-total insensitivity with respect to changes in the parameter regime. This is because the master equation approach does not allow for the formation of hybrid XT-CT states and state delocalisation and thus cannot account for charge separation via the efficient hot exciton dissociation mechanism.

## Effect of interface defect

So far, we have focussed on the dynamics of free charge photo-generation in a perfectly ordered heterojunction, as a crystalline morphology of the donor/acceptor aggregates has proven beneficial to charge generation efficiency[4] and increased the role played by delocalisation[26–37]. Here, we introduce a prototypical interfacial defect into the PDI acceptor phase (see Supplementary Fig. 15), to examine its potential effect on exciton dissociation and the dynamics of the CT-states. We focus on the dynamics of a system where free charges are being mostly generated by hot exciton dissociation (simulation 5 in Table 2), as we expect the hot pathway to be most affected by the introduction of a defect. The change in population dynamics upon the introduction of the defect is displayed in Fig. 4d. Exciton dissociation has slowed down markedly, as the disruption in a6T-PDI stacking at the interface has reduced the coupling between the XT and CT states and reduced the density of hybrid XT-CT states, which are necessary for exciton dissociation (see Fig. 3). This results in a significantly lower

occupation of CT-states throughout the dynamics. Moreover, a smaller fraction of excitons now dissociate via the hot pathway, whilst a more significant fraction dissociate via the cold pathway that is prone to non-radiative recombination. The interfacial defect stifles the hot pathway by disrupting electron transfer between PDI molecules near the interface. This reduces the density of hybrid XT-CT states with charges delocalised further away from the interface, which are instrumental to hot exciton dissociation (see Fig. 5).

## Comparison with other theories

The standard approach for the simulation of charge generation in organic interfaces, KMC employing Marcus rates, has seen success in comparison to experiment[23,25], but this approach can only be applied to electronically weakly coupled systems, e.g. structurally disordered materials, where excitons and charges are localised on molecules or molecular units. It cannot be justifiably applied to ordered, electronically well connected systems where delocalised electronic states are likely to dominate the relaxation dynamics (as shown above). Whilst we have tackled this problem with X-SH non-adiabatic dynamics simulation, Kassal and coworkers took a different path and developed a compelling extension to KMC coined dKMC[32–34]. Starting with a fully quantum mechanical Hamiltonian description in terms of a local electronic basis and harmonic nuclear modes, they utilised the polaron transformation to encode any strong system-bath coupling into the electronic subsystem, and propagate the resulting electronic density matrix via the Redfield[59] equation. By integrating this piecewise and mapping it to KMC, they propagate the evolution of excitation density through transitions between the eigenstates of the Hamiltonian of the full system, thereby accounting for delocalisation.

In their recent application of dKMC, Balzer et al. made qualitative observations that are similar to those reported in our current work[34]. They observed that excitons may dissociate fairly remotely from the interface (thereby directly accessing niCT states) as well as an increased incidence of hybridised states with increasing coupling strength. Their earlier work found that even a marginal increase in carrier delocalisation accelerated charge generation from CT states[33], which they attributed to an increase in overlap between eigenstates involved in charge separation. The qualitative similarities are striking and reassuring given that dKMC and X-SH are based on very different theoretical foundations. An advantage of X-SH is its full atomistic nature allowing for a most realistic simulation of molecule-specific donor-acceptor interfaces that includes all vibrational degrees of freedom explicitly without assuming harmonic interaction potentials and without the need of guessing model parameters, of which there are many in such simulations. Importantly, X-SH encompasses a wide range of exciton transport and dissociation regimes (in contrast to Marcus or Redfield theory) which makes this simulation tool widely applicable, especially to the latest generation of NFAs. The drawback of X-SH is the higher computational cost when compared to, e.g., dKMC, but as we have shown here X-SH simulations on the 10–100 ps timescale of truly nanoscale systems are feasible.

Besides dKMC, fully quantum-dynamical methods have been employed to study charge separation in simplified donor-acceptor interface models. Previous studies have shown that the quantum nature of high-frequency vibrational modes can, in some cases, influence the wavepacket dynamics, as evidenced by coherent beatings in the populations[60]. These coherent oscillations can affect relaxation processes and charge separation on ultrafast timescales. From our previous comparisons with fully quantum-dynamical simulations (using MCTDH for the same electronic Hamiltonian), we found that such initial dynamics are captured with reasonable accuracy by X-SH, even though the high-frequency modes are treated classically. An important benefit of our approach over a fully quantum-mechanical treatment, beyond the significant computational efficiency, is that our scheme correctly recovers the long-time detailed balance, a fundamental requirement for processes occurring on extended timescales where essentially all vibrational modes are in thermal equilibrium and vibronic coherences are significantly damped[48].

## Comparison with experiment

Our simulations reproduce salient experimental observations of the manner of exciton dissociation. This includes the resolution of exciton dissociation into rapid interfacial and diffusion-controlled regimes[14,57], and the initial relaxation of the exciton largely outpacing charge separation[61]. A valuable insight into the charge generation timescales predicted by X-SH can be gleaned through a comparison to the work by Bakulin et al.[13] They utilised a secondary, IR-wavelength pump pulse following a standard excitation pulse to excite charge carriers located in bound CT states at the heterojunction interface, measuring the subsequent increase in photocurrent. Bakulin et al.[13] conducted this measurement on two NFA systems (among other fullerene acceptors), which showed that exciton dissociation occurs on the timescale of several ps, as in several other works[16,17,57,62,63]. They explained the increase in photocurrent through the transient occupancy of higher-energy delocalised CT states that preceded total charge separation, which occurred on the timescale of 10 − 50 ps. Qian et al.[10] also measured the rate of exciton dissociation in a number of low-offset NFA blends, using transient absorption spectroscopy where the resulting signal of the probe pulse was decomposed into excitons and distinct charge carriers. Significant exciton dissociation occurred in most blends within 10 ps. Thus, X-SH can qualitatively reproduce the aforementioned experimental timescales of exciton dissociation and charge separation, with an electronic Hamiltonian solely parameterised with respect to DFT, that explicitly incorporates the effect of fluctuating couplings between electronic states.

As mentioned earlier, we only considered singlet states in this work, disregarding any potential formation of low-lying triplet states. Gillett et al. provided a comprehensive experimental assessment of the role of triplet CT states in low-offset and NFA-based OSCs[64]. They can be close in energy to their singlet counterparts because the exchange energy decreases rapidly with increasing electron-hole separation[65]. Triplet CT states may form via a geminate spin-mixing mechanism from singlet CT states on nanosecond timescales; however, Gillett et al. found no experimental evidence for this geminate pathway in the NFA blends they examined. Instead, they identified a dominant non-geminate mechanism, in which the recombination of free charges generates triplet CT excitons that can undergo fast back charge transfer to Frenkel exciton triplets. However, the non-geminate recombination pathway requiring free charge carriers happens on much longer time scales than the current X-SH simulations can access. Yet, depending on the material under consideration, triplet formation may occur faster, in which case it provides additional entropic driving force for free charge generation[66,67].

## Design rules

Finally, in light of the results obtained from current X-SH simulation, we discuss favourable materials characteristics for organic photovoltaics that may assist experimentalists in selecting suitable combinations of donor and acceptor materials. 1. Small exciton binding energy, $E_b \approx 0$ meV, brought about by, e.g., an increase in dielectric constant to $> 5$. This has also been predicted through experiment[68], but with an emphasis on the prevention of voltage loss. X-SH has shown that in addition to facilitating the escape of electron-hole pairs from energetically low lying iCT states, a reduction in $E_b$ also signifies increased energetic overlap between the lower exciton band edge states and niCT states. 2. Large electronic couplings to generate delocalised CT states, $H_{kl}^{HT} > \lambda_i^{HT}/2$, $H_{kl}^{ET} > \lambda_i^{ET}/2$. It has also been shown experimentally that increased carrier mobility can increase OSC efficiency[69,70]. X-SH finds that an additional effect of the increased CT-bandwidth is an improvement of its overlap with the

exciton band edge resulting in a higher density of delocalised hybrid XT-CT states and a more effective transition from the latter states to niCT and CS states. 3. Large excitonic couplings to generate delocalised XT states, $H_{kl}^{XT} > \lambda_{kl}^{XT}/2$. In addition to faster exciton transport to the interface, this will result in a higher density of delocalised hybrid XT-CT states, similarly as in 2. above, and a more effective transition to these states from pure excitonic states. 4. Structurally ordered donor/acceptor aggregates. The introduction of an interfacial defect that disrupts electron transfer in the PDI acceptor phase near the interface resulted in the cold exciton dissociation pathway gaining significance with respect to the hot pathway, leaving electron-hole pairs more likely to geminately recombine after exciton dissociation. 1.-4. will generally suppress the formation of kinetically trapped iCT states and increase the contribution of hot exciton dissociation. Lastly, a small energy gap between the LUMO and higher lying unoccupied orbitals of the acceptor can further support free charge generation by increasing the entropic advantage of CT vs XT states, as suggested in ref. 71. In the present system the energy gap between LUMO and LUMO+1 was relatively large (1.8 eV), which is why it was not included in our simulations. However, when this gap is small, the entropic advantage is expected to further enhance the beneficial effects of strong electronic coupling and a small exciton binding energy, in comparison to the results reported in this study.

## Outlook

X-SH strikes an optimal balance between theoretical rigour and computational efficiency. An important advantage over other model Hamiltonian approaches is the atomistic resolution of the heterojunction, whose electronic Hamiltonian is parameterised with respect to DFT, and then propagated on-the-fly using ultrafast site energy and electronic coupling estimators. Thermal fluctuations of site energies and electronic couplings, which determine the transport physics of organic semiconductors are naturally included and do not need to be assumed or kept frozen, as is often the case in the literature. This facilitates a realistic and physically intuitive comparison of the dynamics between different OSC materials, as any difference in performance can be directly attributed to electronic parameters intrinsic to the molecular species or its morphology.

The current computational expense of X-SH still limits its practical use to about 40 molecules per donor or acceptor phase due to the quadratic scaling of the size of the CT block with number of molecules. While this is commensurate for the modelling of large 1D or small 2D systems it currently prevents the modelling of 3D systems. This limitation will be addressed in future work. The current X-SH simulations focused on the short time relaxation dynamics upon photo-excitation up to 10 ps. Non-radiative recombination to the ground state has not been included in the present simulations because, according to simple rate-based estimates, the recombination kinetics is expected to occur on significantly longer time scales[2]. However, as the simulation time that can be accessed by X-SH increases, the recombination to the ground state needs to be included such that the competition between charge separation and recombination can be leveraged to yield IQEs for atomistic heterojunction structures.

## Methods

### Excitonic State-based Surface-hopping (X-SH)

As described in the section Results and Discussion, the total electronic Hamiltonian in X-SH is expressed as a sum of Frenkel exciton (XT), charge transfer (CT) and XT-CT coupling Hamiltonians,

$$\hat{H}(\mathbf{R}) = \hat{H}^{XT}(\mathbf{R}) + \hat{H}^{CT}(\mathbf{R}) + \hat{H}^{XT-CT}(\mathbf{R}), \qquad (2)$$

where the dependence on the (classical) nuclear coordinates $\mathbf{R}$ is explicitly stated. The XT block is given by

$$\hat{H}^{XT} = \sum_{k=1}^{M_{XT}} H_{kk}^{XT}(\mathbf{R})|\phi_k^{XT}\rangle\langle\phi_k^{XT}| + \sum_{k=1}^{M_{XT}}\sum_{k\neq l}^{M_{XT}} H_{kl}^{XT}(\mathbf{R})|\phi_k^{XT}\rangle|\langle\phi_l^{XT}| \qquad (3)$$

where $H_{kk}^{XT}(\mathbf{R})$ and $|\phi_k^{XT}\rangle = |\phi_k^{XT}\rangle(\mathbf{R})$ respectively refer to the (quasi-)diabatic site energy and wavefunction of XT state $k$, in which molecule $k$ is in the lowest singlet excited state (Frenkel exciton) and all other molecules are in the ground state, and $H_{kl}^{XT}(\mathbf{R})$, $k \neq l$, refers to the electronic coupling between two diabatic XT states, where the excitons are localised on different molecules $k$ and $l$. Here, the number of exciton states, $M_{XT}$, is equal to the number of acceptor molecules, $M_A$, $M_{XT} = M_A$.

The CT block is given by

$$\hat{H}^{CT} = \sum_{k=1}^{M_D}\sum_{l=1}^{M_A} H_{kl}^{CT}(\mathbf{R})|\phi_{kl}^{CT}\rangle\langle\phi_{kl}^{CT}| + \sum_{k,m=1}^{M_D}\sum_{l,n=1}^{M_A\prime} H_{kl,mn}^{CT}(\mathbf{R})|\phi_{kl}^{CT}\rangle\langle\phi_{mn}^{CT}| \qquad (4)$$

where $H_{kl}^{CT}(\mathbf{R})$ and $|\phi_{kl}^{CT}\rangle(\mathbf{R})$ refer to the diabatic site energy and wavefunction of CT state $kl$ with the excess hole on molecule $k$ of the $M_D$ donor molecules and the excess electron on molecule $l$ of the $M_A$ acceptor molecules, and $H_{kl,mn}^{CT}(\mathbf{R})$ refers to the electronic coupling between CT states $kl$ and $mn$. The total number of CT states is then $M_{CT} = M_A M_D$. The primes on the right hand side of Eq. (4) indicates that the diagonal term ($k = m, l = n$) is excluded. Lastly, the XT-CT block contains the electronic couplings between XT and CT states,

$$
\begin{aligned}
\hat{H}^{XT-CT} = & \sum_{k=1}^{M_{XT}}\sum_{m=1}^{M_D}\sum_{n=1}^{M_A} H_{k,mn}^{XT-CT}(\mathbf{R})|\phi_k^{XT}\rangle\langle\phi_{mn}^{CT}| \\
& + \sum_{m=1}^{M_D}\sum_{n=1}^{M_A}\sum_{k=1}^{M_{XT}} H_{mn,k}^{XT-CT}(\mathbf{R})|\phi_{mn}^{CT}\rangle\langle\phi_k^{XT}|
\end{aligned} \qquad (5)
$$

Insertion of the electronic wavefunction

$$\Psi(t) = \sum_{k=1}^{M_{XT}} u_k^{XT}(t)|\phi_k^{XT}\rangle + \sum_{m=1}^{M_D}\sum_{n=1}^{M_A} u_{mn}^{CT}(t)|\phi_{mn}^{CT}\rangle \equiv \sum_{i=1}^{M} u_i(t)|\phi_i\rangle \qquad (6)$$

into the time-dependent Schrödinger equation, gives the time evolution of the electronic states in terms of the diabatic expansion coefficients $u_i$,

$$\frac{du_i(t)}{dt} = \sum_{j=1}^{M} u_j(t)(H_{ij}(\mathbf{R}) - i\hbar d_{ij}), \qquad (7)$$

where for ease of notation we have introduced $u_i$ and $|\phi_i\rangle$, $u_i \in \{\{u_k^{XT}\}, \{u_{mn}^{CT}\}\}$ and $|\phi_i\rangle \in \{\{|\phi_k^{XT}\rangle\}, \{|\phi_{mn}^{CT}\rangle\}\}$, $k = 1,...,M_{XT}$, $m = 1,...,M_D$, $n = 1,...,M_A$, $i = 1,...,M$, $M = M_{XT} + M_D M_A$. $H_{ij}$ denotes the matrix elements of the total electronic Hamiltonian (Eq. (2)); the non-adiabatic coupling elements between the quasi-diabatic states, $d_{ij} = \langle\phi_i|\frac{d}{dt}|\phi_j\rangle$, are generally small (see ref. 40 for a discussion) and are neglected.

In surface hopping, the nuclear force on only a single active adiabatic electronic state (index $a$) is required, $\mathbf{F}_{I,a}$, which is given by ref. 72

$$\mathbf{F}_{I,a} = -[\mathbb{U}^\dagger(\nabla_I\mathbb{H})\mathbb{U}]_{aa} \qquad (8)$$

where $\nabla_I = \frac{\partial}{\partial\mathbf{R}_I}$, $\mathbb{H}$ is the diabatic electronic Hamiltonian Eq. (2) in matrix form, and $\mathbb{U}$ is a unitary transformation diagonalizing $\mathbb{H}$. Each MD time step, the probabilities for a surface hop from the active electronic state $a$ to the other adiabatic states $j$ are calculated

according to Tully's expression[73], which contains the non-adiabatic coupling element (NACE) between the adiabatic electronic states, $d_{ja}^{\text{ad}}$. The NACEs are obtained through[74] $d_{ja}^{\text{ad}} = \langle \psi_j | \dot{\psi}_a \rangle = [\mathbb{U}^\dagger \mathbb{D} \mathbb{U}]_{ja} + [\mathbb{U} \dot{\mathbb{U}}]_{ja} \approx [\mathbb{U} \dot{\mathbb{U}}]_{ja}$, where the NACEs between (quasi-)diabatic states ($d_{kl} \equiv [\mathbb{D}]_{kl}$) are neglected. Energy conservation is ensured by only allowing trajectories to hop if their kinetic energy along the direction of the non-adiabatic coupling vector (NACV) is equal to or larger than the difference between the two potential energy surfaces ($E_j - E_a$), with the nuclear velocities being reversed in the direction of the NACV if the hop is rejected[74,75]. We adopt the same modifications to the standard FSSH algorithm as in our previous works[76] to ensure internal consistency and (approximate) detailed balance in addition to total energy conservation. This includes an energy-based decoherence correction[77], a state-tracking algorithm to prevent trivial crossings, and the elimination of spurious long-range exciton and charge transfer. We refer to ref. 76 for a detailed discussion of these modifications and their physical underpinnings.

## Site energies

In practice, we define all site energies with respect to the lowest energy state of the electronic Hamiltonian, which is the iCT state $|\phi_{11}^{\text{CT}}\rangle$. Notice that the electronic ground state (S0) is not included in our model because it will be populated on time scales that are much longer than the present X-SH simulations. The site energy of XT states is then given by

$$H_{kk}^{\text{XT}}(\mathbf{R}) = \epsilon^{\text{XT}} + \Delta E_k^{\text{XT}}(\mathbf{R}), \qquad (9)$$

where $\epsilon^{\text{XT}}$ is the electronic energy of the Frenkel exciton at the interface $|\phi_1^{\text{XT}}\rangle$ relative to the one for the iCT state $|\phi_{11}^{\text{CT}}\rangle$ in the minimum energy configuration of the electronic ground state, $\mathbf{R}_{\text{GS}}$, $\epsilon^{\text{XT}} = E_1^{\text{XT}}(\mathbf{R}_{\text{GS}}) - E_{11}^{\text{XT}}(\mathbf{R}_{\text{GS}})$. $\epsilon^{\text{XT}}$ is assumed to be the same for all Frenkel excitons $k$. $\Delta E_k^{\text{XT}}(\mathbf{R})$ is the electronic energy of $|\phi_k^{\text{XT}}\rangle$ at a nuclear configuration $\mathbf{R}$ relative to the energy of that state at $\mathbf{R}_{\text{GS}}$, $\Delta E_k^{\text{XT}}(\mathbf{R}) = E_k^{\text{XT}}(\mathbf{R}) - E_k^{\text{XT}}(\mathbf{R}_{\text{GS}})$. The latter term takes into account the thermal fluctuations of the site energy related to the local electron-phonon coupling. $\epsilon^{\text{XT}}$ is calculated for a gas-phase interfacial a6T:PDI dimer, whose geometry was extracted from the crystal structure of the heterojunction. The adiabatic excited state energies of the dimer were first calculated with TDDFT, and the energy difference between the localised PDI exciton and interfacial CT-state was subsequently obtained with the MS-FED-FCD[78,79] diabatisation procedure. This gave $\epsilon^{\text{XT}} = 390$ meV. For calculation of $E_k^{\text{XT}}(\mathbf{R})$, the electronic ground state potential energy for a6T molecules and for PDI molecules $l \neq k$ were described by the standard Generalised Amber Force Field (GAFF)[80], while for the excited PDI molecule $k$ the equilibrium bond lengths were adjusted to reproduce the internal reorganisation energy for S1 exciton transfer between two PDI molecules from (TD)DFT calculations, $\lambda_i^S$, $S = $ XT. All other parameters for the excited PDI molecule $k$ remain the same as in the electronic ground state. The internal reorganisation energy was calculated in the gas phase using the 4-point scheme:

$$\lambda_i^S = [E_A(\mathbf{R}_B) + E_B(\mathbf{R}_A)] - [E_A(\mathbf{R}_A) + E_B(\mathbf{R}_B)] \qquad (10)$$

where for exciton transfer ($S = $ XT), $E_A$ and $E_B$ denote the potential energy of the S1 excited state ($A = $ S1) and S0 ground state ($B = $ S0) of a PDI molecule in vacuum, respectively, and $\mathbf{R}_A$ and $\mathbf{R}_B$ are the nuclear coordinates at the minimum of the S1 and S0 potential energy surfaces, respectively. The ground and excited state calculations for $\epsilon^{\text{XT}}$ and $\lambda_i^{\text{XT}}$ were carried out using the CAM-B3LYP[81] functional and the 6-31G(d,p) basis set as implemented in the Gaussian16[82] electronic structure software package.

The site energy of CT states is given by

$$H_{kl}^{\text{CT}}(\mathbf{R}) = -\frac{1}{\epsilon_{\text{r}}} \left( \frac{1}{r_{kl}} - \frac{1}{r_{11}} \right) - |\mathbf{E}|(r_{kl} - r_{11}) + \Delta E_{kl}^{\text{CT}}(\mathbf{R}) \qquad (11)$$

where the first term on the right hand side of Eq. (11) describes the Coulomb interaction between the electron-hole pair located on PDI molecule $l$ and a6T molecule $k$ at configuration $\mathbf{R}_{\text{GS}}$, relative to the Coulomb interaction of the iCT state ($l = 1$, $k = 1$) at configuration $\mathbf{R}_{\text{GS}}$. $r_{kl}$ is the electron-hole distance between the centres of mass of the respective molecules and $\epsilon_{\text{r}}$ is the dielectric constant. The second term on the right hand side of Eq. (11) describes the interaction of the electron and hole with a static electric field in the direction orthogonal to the interface and of magnitude equal to $|\mathbf{E}|$, relative to the electric field interaction in the iCT state, again taken at configuration $\mathbf{R}_{\text{GS}}$. The field term is introduced to account for the bias due to the presence of external electrodes in OSC devices. The magnitude of the electric field was set to $10^5$ V cm$^{-1}$, a value that is typical for OSC devices[20,52]. The third term on the right hand side of Eq. (11) accounts for thermal fluctuations of the site energies and is defined as the electronic energy of state $|\phi_{kl}^{\text{CT}}\rangle$ at a nuclear configuration $\mathbf{R}$ relative to the energy of that state at $\mathbf{R}_{\text{GS}}$. $\Delta E_{kl}^{\text{CT}}(\mathbf{R}) = E_{kl}^{\text{CT}}(\mathbf{R}) - E_{kl}^{\text{CT}}(\mathbf{R}_{\text{GS}})$. Note that since the iCT state is used as a baseline, $H_{11}^{\text{CT}}(\mathbf{R}) = \Delta E_{11}^{\text{CT}}(\mathbf{R})$ throughout X-SH dynamics. Analogous to XT states, for the calculation of $E_{kl}^{\text{CT}}(\mathbf{R})$, the electronic ground state potential energy for a6T molecules and PDI molecules ($m \neq k$, $n \neq l$) was described by the standard Generalised Amber Force Field (GAFF)[80], while for the positively charged a6T molecule $k$ and the negatively charged PDI molecule $l$ the respective equilibrium bond lengths were adjusted to reproduce the internal reorganisation energy for hole transfer (HT) between two a6T and electron transfer (ET) between two PDI molecules from DFT calculations, respectively. All other parameters for the charged a6T and PDI molecules remain the same as in the neutral electronic ground state. As for the XT state, the internal reorganisation energy (subscript i) for HT and ET was calculated in the gas phase using the 4-point scheme (Eq. (10)) where $S = $ HT, ET. Here, $E_A$ and $E_B$ denote the potential energy of the ground state for the charged molecule ($A = $ C) and neutral molecule ($B = $ N) in vacuum, respectively, and $\mathbf{R}_A$ and $\mathbf{R}_B$ are the nuclear coordinates at the minimum of the potential energy surfaces in the C and N state, respectively. The DFT calculations for $\lambda_i^{\text{HT}}$ and $\lambda_i^{\text{ET}}$ were carried out in CP2K[83] with the B3LYP functional[84] and the 6-31G(d,p) basis set. The thermal fluctuations of the first two terms in Eq. (11) (which are part of $\Delta E_{kl}^{\text{CT}}(\mathbf{R})$) are very small due to negligible centre of mass motion ($r_{kl}$) of the molecules in our assembly and they are thus neglected.

The nuclear gradients of the site energies, required for the calculation of nuclear forces on the adiabatic electronic states (Eq. (8)) and NACVs, are obtained from the force field. Note that in Eqs. (9) and (11) only the electron-phonon terms, $\Delta E_k^{\text{XT}}(\mathbf{R})$ and $\Delta E_{kl}^{\text{CT}}(\mathbf{R})$, respectively, depend on nuclear coordinates whereas the electronic terms are coordinate independent.

## Coupling matrix elements

Excitonic couplings between XT states are calculated using the transition charge approach[54],

$$H_{kl}^{\text{XT}}(\mathbf{R}) = \sum_{K \in k} \sum_{L \in l} \frac{q_K^{\text{T}} q_L^{\text{T}}}{|\mathbf{R}_K - \mathbf{R}_L|} \qquad (12)$$

where $K$ and $L$ label atoms in PDI molecules $k$ and $l$, respectively, $q_K^{\text{T}}$ and $q_L^{\text{T}}$ are the TrESP transition charges and $\mathbf{R}_K$, $\mathbf{R}_L$ the atomic positions. The TrESP charges for PDI were taken from our previous work[40] where they were obtained from CAM-B3LYP calculations in Gaussian using the 6−31G(d,p) basis set. The transition charges were frozen during X-SH dynamics. We have shown in previous work that the TrESP approach gives very accurate results when compared to full excitonic

couplings (that also include exchange and overlap contributions). Moreover, it was shown that the small fluctuations of transition charges during dynamics can be safely neglected. The nuclear gradients of the excitonic couplings, which are required for the calculation of the nuclear force on the adiabatic electronic states (Eq. (8)) and for NACVs, are obtained analytically from Eq. (12).

As to the calculation of electronic coupling between CT states, we note that in a given CT state $|\phi_{kl}^{CT}\rangle$, the hole and electron are located on distinct molecules, a6T molecule $k$ and PDI molecule $l$, respectively. Hence, we assert that the CT-state can be written as a single product of localised electron and hole wavefunctions, $|\phi_{kl}^{CT}\rangle \simeq |\phi_k^h\rangle|\phi_l^e\rangle$. The coupling matrix elements between two CT states $|\phi_{kl}^{CT}\rangle$ and $|\phi_{mn}^{CT}\rangle$ are then approximated as follows,

$$H_{kl,mn}^{CT} = \langle \phi_{kl}^{CT}|\hat{H}^{CT}|\phi_{mn}^{CT}\rangle \qquad (13)$$

$$\simeq \langle \phi_k^h \phi_l^e|\hat{H}^{CT}|\phi_m^h \phi_n^e\rangle \qquad (14)$$

$$\simeq \langle \phi_k^h|\hat{H}^h|\phi_m^h\rangle\delta_{ln}(1-\delta_{km}) + \langle \phi_l^e|\hat{H}^e|\phi_n^e\rangle\delta_{km}(1-\delta_{ln}) \qquad (15)$$

$$= H_{km}^{HT}\delta_{ln}(1-\delta_{km}) + H_{ln}^{ET}\delta_{km}(1-\delta_{ln}) \qquad (16)$$

According to Eq. (15) only those CT states couple that differ in the occupation of either the hole ($k \neq m$, $l = n$) or the electron ($k = m$, $l \neq n$) but not in the occupation of both particles. The matrix element between the latter CT states ($k \neq m$, $l \neq n$) is very small compared to the former cases and is neglected[85]. The integration yields the electronic Hamiltonian of the donor phase hosting the hole, $\hat{H}^h$, and the electronic Hamiltonian of the acceptor phase hosting the electron, $\hat{H}^e$. The electronic couplings for hole transfer in the donor phase, $H_{km}^{HT}$, and for electron transfer in the acceptor phase, $H_{ln}^{ET}$, are calculated using the ultrafast Analytic Overlap Method (AOM)[55,56]. In AOM, electronic couplings are estimated via the linear relationship $H_{ab} = \bar{C}\bar{S}_{ab}$ where $\bar{S}_{ab}$ is the orbital overlap between the relevant frontier orbitals of electron-donating and electron-accepting molecules in vacuo (HOMO-HOMO for hole transfer, LUMO-LUMO for electron transfer) and $\bar{C}$ is a constant of proportion determined by fitting the above linear relationship to electronic coupling values obtained from DFT calculations and projection operator-based diabatisation (POD(PBE), 6-31G(d,p))[86].

AOM is highly efficient because $\bar{S}_{ab}$ is calculated analytically along the non-adiabatic dynamics trajectories for each molecular pair using frozen orbital expansion coefficients, see ref. 26. This method has been extensively benchmarked in our previous works and was found to give good accuracy on a test set from the HAB79 data set[87] with a mean relative unsigned error of 24.7% (for the range of couplings relevant to this work (10–100 meV)), while reducing the computational cost by 4–5 orders of magnitude compared with DFT(POD) calculations[56]. The scaling constants $\bar{C}^{HT}$ and $\bar{C}^{ET}$ were taken to be the scaling constant (9.463 and $-9.463$ eV, respectively) obtained from fitting against the HAB79 dataset[87]. The goodness of fit was tested by sampling 100 configurations along a 1ps X-SH run at 300 K and comparing the resultant AOM couplings against DFT(POD) calculations. Both couplings displayed good agreement between POD and AOM for the chosen constant ($R^2$ = 0.904 for $\bar{C}^{HT}$ and $R^2$ = 0.963 for $\bar{C}^{ET}$, Supplementary Fig. 14).

The electronic coupling between XT and CT-states, $H_{k,mn}^{XT-CT}$, is limited to the donor-acceptor molecular pair at the interface owing to its fast exponential decay. All other elements in the $\hat{H}^{XT-CT}$ block are also calculated at each timestep, but are either close or equal to zero. The XT-CT coupling corresponds to an excited state hole transfer from the interfacial S1 PDI molecule to the interfacial a6T

molecule and can thus be estimated using AOM, similarly to the procedure described above. The constant of proportion, $\bar{C}^{XT-CT}$, is obtained by fitting a linear relationship between the overlap of the HOMO orbitals of interfacial a6T and PDI to excited state electronic coupling values obtained from TDDFT calculations and MS-FED-FCD diabatisation[78,79] (CAM-B3LYP, 6-31G(d,p)). A fit on 25 configurations sampled along a 1ps X-SH trajectory at 300 K gives a value $\bar{C}^{XT-CT}$ = 17.06 eV ($R^2$ = 0.618), see Supplementary Fig. 14. The nuclear gradients of all electronic coupling matrix elements are obtained numerically via finite differences using AOM. We finally note that the signs of the off-diagonal elements in the electronic Hamiltonian are, by construction, consistent with the phases of the corresponding molecular orbitals, which are in-turn dependent on the atomic ordering of such molecules. We refer to ref. 38 for a detailed discussion.

### Simulation protocol

An atomistic model for the a6T:PDI interface is constructed from the experimental crystal structures of a6T and PDI. Our aim here is to build a plausible and thermally stable interface structure where molecules adopt the single crystal packing structure some distance away from the interface. To this end, initial atomic coordinates of the a6T and PDI single crystals were taken from the Cambridge Crystallographic Data Centre with structural identifiers ZAQZUM01 and LENPEZ01, respectively. The PDI supercell (containing 240 molecules) was constructed from $20 \times 6 \times 1$ unit cells, whilst the a6T supercell (300 molecules) was constructed from $10 \times 15 \times 1$ unit cells. The interface was then created by the manual alignment of each supercell along the $b$ and $c$-directions, followed by their combination along their respective $a$-directions, such that the $\pi - \pi$ stacking between interfacial molecules is maximised, resulting in a distance between the centers of mass of interfacial a6T and PDI molecules of 9.8 Å. Periodic boundary conditions were then applied along the $b$ and $c$-directions using lattice constants that are very close to the ones for the respective pristine crystals thereby compensating for the small lattice mismatch. To preserve the heterojunction structure along the non-periodic $a$-direction, we applied weak restraints to the geometric centres of only the upper/lower rings of each molecule, in both the equilibration and production runs. The dimensions of the simulation cell were $300 \times 300 \times 300$ Å$^3$ and kept constant during equilibration and production runs. The initial interface structure was equilibrated for 500 ps to 300 K in the electronic ground state (GS) using a Nosé-Hoover thermostat. Whilst the orientations of the interfacial molecules slightly deviate from the one in the pristine crystal, the molecules further apart adopt the orientation of the respective pristine crystal as indicated by a low RMSE of 0.228 Å with respect to the experimental crystal structures. To proceed, 1000 nuclear configurations were equidistantly sampled along a subsequent 1 ns run in the NVE ensemble and used as initial geometries for X-SH. We note in passing that the generated structure may not be the energetically most stable interface structure - more sophisticated structure generation tools would be required to look at different interface structures but this is beyond the intended scope of the current work.

X-SH simulations were carried out for 7 different parameter regimes summarised in Table 2. The electronically active region in which XT and CT states are propagated was chosen to be a one-dimensional chain of 20 a6T and 20 PDI molecules along the $a$-direction of the above equilibrated interface. All other molecules in the supercell were treated as electronically inactive and interacted with the active region only via non-bonded interactions. Photo-excitation was modelled by a 10 fs Gaussian laser pulse. Each X-SH trajectory was initiated in an excited eigenstate randomly taken from a distribution of eigenstates with a total probability that is proportional to the probability for the incoming photon energy being equal to the energy gap between the ground and excited eigenstate, the corresponding

transition dipole moment and the probability of the excited eigenstate being sampled along an electronic ground state MD trajectory of the a6T:PDI interface at 300 K. The mean photon energy of the laser pulse for each parameter regime was selected to maximally overlap with the region of the XT band (obtained by sampling ground-state MD) with the highest transition dipole moment. We refer to Supplementary Discussion 2 for full details on initial excited state preparation. In each of the 500 X-SH trajectories, the chosen initial electronic eigenstate was assigned the initial active adiabatic state, $\psi_a(t = 0)$, and taken to be the initial electronic wavefunction, $\Psi(t = 0) = \psi_a(t = 0)$. The trajectories were propagated in the NVE ensemble for 10 ps using a nuclear timestep $\Delta t = 0.05$ fs. The electronic equation (7) was integrated with the 4th order Runge-Kutta algorithm (RK4), using an electronic timestep $\delta t = 0.01$ fs. All X-SH simulations were carried out with our in-house implementation of X-SH in the CP2K simulation package[83].

## Property calculation

The XT, iCT, niCT and CS populations shown in Fig. 4 are calculated as

$$P_{\text{XT}} = \left\langle \sum_{k=1}^{M_{\text{XT}}} |u_k^{\text{XT}}(t)|^2 \right\rangle_N \tag{17}$$

$$P_{\text{iCT}} = \left\langle |u_{11}^{\text{CT}}(t)|^2 \right\rangle_N \tag{18}$$

$$P_{\text{niCT}} = \left\langle \sum_{k=1}^{M_{\text{D}}} \sum_{l=1}^{M_{\text{A}}} H(r_{kl} - r_{11}) H(r_{\max} - r_{kl}) |u_{kl}^{\text{CT}}(t)|^2 \right\rangle_N \tag{19}$$

$$P_{\text{CS}} = \left\langle \sum_{k=1}^{M_{\text{D}}} \sum_{l=1}^{M_{\text{A}}} H(r_{kl} - r_{\max}) |u_{kl}^{\text{CT}}(t)|^2 \right\rangle_N \tag{20}$$

where $H(x)$ is the Heaviside step function modified such that $H(x) = 0$ when $x \leq 0$, $r_{11}$ is the centre of mass distance between the interfacial a6T-PDI molecular pair, $r_{\max}$ is the distance at which the Coulomb potential (first two right-hand-side terms of Eq. (11)) is at its maximum, and $\langle ... \rangle_N$ denotes the average over $N = 500$ X-SH trajectories.

The inverse participation ratio (IPR) is a measure of the wavefunction delocalisation and is usually defined as IPR $= 1/(\sum_k |u_k|^4)$, where $u_k$ are diabatic expansion coefficients. The presence of different particle-types in X-SH (XT and CT) precludes the direct use of this definition because the squares of the diabatic expansion coefficients of a given particle type no longer sum to unity, which underestimates the purity of the total electronic wavefunction $\Psi(t)$ with respect to each particle type. Thus we first renormalise the XT contribution to $\Psi(t)$ and insert this in the usual definition for IPR, giving

$$\text{IPR}_{\text{XT}}(t) = \frac{\left( \sum_{l=1}^{M_{\text{XT}}} |u_l^{\text{XT}}(t)|^2 \right)^2}{\sum_{k=1}^{M_{\text{XT}}} |u_k^{\text{XT}}(t)|^4}. \tag{21}$$

To calculate the IPR of the electron in the acceptor phase, we first obtain the probability of an electron being found on an acceptor molecule $l$ and the hole on any donor molecule $k$, $\sum_{k=1}^{M_{\text{D}}} |u_{kl}(t)|^2$, and then renormalise the CT contribution to $\Psi(t)$ giving

$$\text{IPR}_{\text{e}}(t) = \frac{\left( 1 - \sum_{m=1}^{M_{\text{XT}}} |u_m^{\text{XT}}(t)|^2 \right)^2}{\sum_{l=1}^{M_{\text{A}}} \left( \sum_{k=1}^{M_{\text{D}}} |u_{kl}^{\text{CT}}(t)|^2 \right)^2}. \tag{22}$$

The IPR of the hole in the donor phase is calculated similarly,

$$\text{IPR}_{\text{h}}(t) = \frac{\left( 1 - \sum_{m=1}^{M_{\text{XT}}} |u_m^{\text{XT}}(t)|^2 \right)^2}{\sum_{k=1}^{M_{\text{D}}} \left( \sum_{l=1}^{M_{\text{A}}} |u_{kl}^{\text{CT}}(t)|^2 \right)^2}. \tag{23}$$

The exciton binding energy is given by:

$$E_{\text{b}} = E_{\text{f}} - E_{\text{g}}^{\text{opt}} \tag{24}$$

where $E_{\text{f}}$ refers to the energy of the uncorrelated electron-hole pair and $E_{\text{g}}^{\text{opt}}$ refers to the vertical excitation energy between the ground state and the lower exciton (S1) band edge. Both quantities in Eq. (24) are calculated relative to the iCT state energy, where $E_{\text{f}}$ naturally varies with respect to the Coulomb barrier height. Further details about how each quantity is calculated are included in Supplementary Discussion 3.

## Data availability

The processed simulation data plotted in this study and the simulation input files have been deposited in the Figshare database [identifier: 10.6084/m9.figshare.30738962][88]. The raw data produced by X-SH has not been shared to a public repository due to size constraints, and is available upon request by contacting Filip Ivanovic or Jochen Blumberger. The expected time frame of a response is 1–5 days, and the data will be made available for one month.

## Code availability

The latest tested version of the custom code used for this study has been placed in a public Zenodo repository with the identifier: https://doi.org/10.5281/zenodo.17792822[89].

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

## Acknowledgements

F.I. was supported by an EPSRC DTP Ph.D. studentship (EP/W524335/1). S.G. gratefully acknowledges funding from Fundings projects presented by young researchers Mission 4 Component 2 Investment line 1.2 (CUP: I53C24002040006) by the European Union-NextGenerationEU-PNRR. J.B. would like to thank the ERC for selecting the ERC Advanced Grant EXCITING and the UKRI for funding it under the Horizon Europe Guarantee (EP/Z533932/1). Via our membership of the UK's HEC Materials Chemistry Consortium, which is funded by EPSRC (EP/L000202, EP/R029431), this work used the ARCHER2 UK National Supercomputing Service (http://www.archer2.ac.uk) as well as the UK Materials and Molecular Modelling (MMM) Hub, which is partially funded by EPSRC (EP/P020194), for computational resources. We also acknowledge the

UCL Myriad high performance computing facility. W.-T.P. acknowledges the financial support from the National Science and Technology Council (NSTC), Taiwan under grant number: 114-2113-M-029-007-MY3.

## Author contributions

F.I., S.G., and W.-T.P. implemented and tested the X-SH code, with input from J.B. S.G. and F.I. parameterised the electronic Hamiltonian, and S.G. created the atomistic heterojunction structure. F.I. performed all of the X-SH simulations, analysis, and additional quantum chemistry calculations, with input from J.B. and S. G. J.B., W.-T. P., and S.G. contributed to the development of the theoretical framework. J.B. contributed to data interpretation, and he supervised all aspects of the research. F.I. and J.B. wrote the manuscript. All authors reviewed and discussed the manuscript.

## Competing interests

The authors declare no competing interests.
