## [Transparent Peer Review file · Nature Communications]

Transiently Delocalised Hybrid Quantum States are Gateways for Efficient Exciton Dissociation at Organic Donor-Acceptor Interfaces

Corresponding Author: Professor Jochen Blumberger

Version 0:

Reviewer comments:

Reviewer #1

(Remarks to the Author)

This is an interesting theoretical study that is appropriate for publication in Nature Communications after the authors consider my minor comments about the context of the work. The paper, by studying parameters in a theoretical model (electronic coupling, dielectric constant), shows convincingly how exciton dissociation is much more efficient when pathways that avoid the bound electron-hole pair state are exploited. This is an important design principle, as noted in the paper.

The paper is appropriate for this journal because the topic is at the current forefront of the field. For example, transient delocalization of excitons has been posited as a new design principle to aid exciton transport:
<https://pubs.acs.org/doi/10.1021/acs.jpcclett.2c01133>

The work, while new and novel in terms of the approach, does echo prior studies, which the authors may consider noting more explicitly to support their motivation and conclusions. Specifically, it is well recognized that the interfacial charge-separated state is a kinetic trap. In this paper
<https://pubs.acs.org/doi/10.1021/nn700179k>
the basis for a band of CT states that provide pathways with diminished exciton binding is presented. That picture is close to the basic starting point for the present study. The present study significantly expands the scope by predicting dynamics. This well-known paper
<https://www.science.org/doi/10.1126/science.1246249>
gave experimental evidence for the advantages of initial charge separation producing a delocalized anion state in the fullerene domain. This paper
<https://pubs.acs.org/doi/abs/10.1021/jp5098102>
is closely related to the present study and points out the entropic advantage of a large density of CT states for preventing recombination.

(Remarks on code availability)

Reviewer #2

(Remarks to the Author)

The authors present a detailed theoretical study of nanoscale donor-acceptor interfaces in a prototypical a6T:PDI system, using the X-SH framework to simulate non-adiabatic molecular dynamics. The results indicate that efficient dissociation of excitonic states into free carriers occurs via transiently delocalized hybrid exciton-charge transfer states. Strong electronic couplings between the donor and acceptor molecules promote 'hot exciton' dissociation, a similar effect to increasing the dielectric constant of the heterojunction. The authors also discuss the experimental implications of their findings. Additionally, the article introduces an improved X-SH methodology (accounting for off-diagonal disorder) that could benefit

future applications to various materials. Given the significance of these findings, I recommend this work for publication in Nature Communications following a revision addressing the comments below. No further review is necessary.

1. The authors consider for their simulations idealized crystalline interfaces between donor and acceptor phases. Indeed, while experimentally it was found that crystalline interfaces work better than amorphous ones for charge separation efficiency, I wonder what the role of defects is. To what extent some typical defects, e.g. faults in the stack of both materials, affect the efficiency of charge transfer mechanisms. These modification can be easily incorporated and tested on the level of X-SH model. This discussion would be an important addition to the paper and point to experimentalists how defect tolerant the charge separation is, for example, for a hot-exciton pathway.

2. There are lots of experimental and theoretical studies in the field pointing to the important of vibronic coherence that can fundamentally affect many photoinduced processes and excited state dynamics. The authors use surface hopping dynamics which fundamentally lacks vibrational wave functions, and decoherence corrections are just fixing the non-adiabatic algorithms. The authors need to include a brief discussion on this phenomenon (which to me could be very important for hot exciton dissociation pathway) and how it can alter the emerging picture.

3. Cold exciton dissociation. Is it correct to assume that the simulations are limited by 10ps of simulations time. Within this timescale, there was no observed dissociation of the iCT states. The subsequent dynamics (i.e. "charge separation or recombination of iCT state") involves longer timescales and this dynamics is not included in the kinetic models presented. Perhaps some clarifying remarks would be helpful here.

4. Hot exciton dissociation. I would question validity of the statement "a small increase in dielectric constant from 3.5 to 5 (Table 2, simulation 4) results in a decrease of exciton binding energy from 260 to 10 meV." This is a huge claim and I do not think that just referring to Eq. 24 provides justice. Exciton binding energy is a delicate quantity to calculate. I would request the authors to substantiate their claims. Experimental evidence is indirect (ref. 61).

5. Regarding the abstract statement: "Our results highlight the importance of electronic couplings, an often overlooked property, for opto-electronic charge generation." I believe that this is not true and tuning of electronic coupling has been an active subject of the discussion in the community for decades across both theoretical and experimental studies. Perhaps this statement and similar tone in the main body of the article need to be softened.

6. The present approach considers singlet states only. However, I would argue that weakly bound triplet charge transfer states are iso-energetic with their singlet counterparts and can be easily form even in the absence of strong spin-orbit coupling. Their presence and participation in the process can add additional entropic driving force further improving charge separation. I would suggest to add discussion of this scenario.

7. Code availability statement says: "The custom codes used for this study are available from the corresponding authors upon request." It would be really helpful to clarify releases of the code for public, for example, if any of the standard version of X-SH are available and what are the plans for realizing the improved version. This is an important advance of this study.

(Remarks on code availability)

I do not think that X-SH code used for this study was released, see my point #7

Version 1:

Reviewer comments:

Reviewer #1

(Remarks to the Author)

The authors have thoughtfully revised their paper, which was already excellent. I enthusiastically recommend acceptance.

(Remarks on code availability)

Reviewer #2

(Remarks to the Author)

The authors have thoroughly addressed all reviewer's comments, with clear attention to detail and a deep understanding of the subject matter. Their responses are thoughtful, comprehensive, and convincing, and the corresponding revisions in the manuscript significantly strengthen the work. Importantly, the authors have also released the relevant code, further supporting the transparency and reproducibility of their results. Given these efforts, I strongly recommend publication of the manuscript in Nature Communications in its present form.

(Remarks on code availability)

RESPONSES TO REVIEWER COMMENTS

This is a document detailing our point-by-point response to each comments by the reviewers. Text in black corresponds to the reviewers' comments repeated verbatim. Blue text corresponds to our responses to these comments, while red text corresponds to adjustments that were accordingly made in the manuscript.

Reviewer #1 (Remarks to the Author):

This is an interesting theoretical study that is appropriate for publication in Nature Communications after the authors consider my minor comments about the context of the work. The paper, by studying parameters in a theoretical model (electronic coupling, dielectric constant), shows convincingly how exciton dissociation is much more efficient when pathways that avoid the bound electron-hole pair state are exploited. This is an important design principle, as noted in the paper.

We thank the reviewer for the very positive remarks on our work and for providing additional context to our simulations.

The paper is appropriate for this journal because the topic is at the current forefront of the field. For example, transient delocalization of excitons has been posited as a new design principle to aid exciton transport:

<https://pubs.acs.org/doi/10.1021/acs.jpcclett.2c01133>

This work has been cited in the original manuscript (Ref. 37).

1. The work, while new and novel in terms of the approach, does echo prior studies, which the authors may consider noting more explicitly to support their motivation and conclusions. Specifically, it is well recognized that the interfacial charge-separated state is a kinetic trap. In this paper

<https://pubs.acs.org/doi/10.1021/nn700179k>

the basis for a band of CT states that provide pathways with diminished exciton binding is presented. That picture is close to the basic starting point for the present study. The present study significantly expands the scope by predicting dynamics.

The suggested paper considers a static Hamiltonian in the basis of localised Frenkel exciton and charge-transfer states and indeed highlights the possible advantage of the energetic alignment of the exciton and CT bands, due to a complementary combination of the strength of Coulombic electron-hole interactions and electronic couplings. However, as the reviewer pointed out, it is a purely static perspective and lacks the important quantum dynamics that we provide in our paper. We have therefore expanded our *Introduction* section on page 3:

“In this respect, a study by Scholes considered the energetics of delocalised eigenstates of a static site-based Hamiltonian, and concluded that the energetic alignment of the exciton and niCT states promotes free charge generation, and avoids the occupation of the kinetically trapped iCT state.[43] Further support for this finding requires the prediction of dynamics, specifically the propagation of time-dependent populations of the electronic states involved in charge generation, which has since been attempted with a variety of methods with varying degrees of accuracy. Gelinias *et al.* modelled their experimental evidence of ultrafast charge separation in a fullerene acceptor heterojunction by propagating the electronic state populations of a similar site-based Hamiltonian with a truncated state space, and concluded

that charge generation in such a blend is assisted by the transient occupation of high-energy niCT states with a delocalised anion.[14] A notable development was made by Vukimirovic and Jankovic [44], who designed an approach to describe the time-dependent population dynamics of delocalised electronic states, starting from a Hamiltonian constructed from the full manifold of localised singlet exciton/CT-states in a model heterojunction.”

2. This well-known paper <https://www.science.org/doi/10.1126/science.1246249> gave experimental evidence for the advantages of initial charge separation producing a delocalized anion state in the fullerene domain.

We thank the reviewer for this reference. This article was referred to in the original manuscript when we referenced experimental evidence of the hot exciton dissociation pathway. The reviewer is however correct that the article should be mentioned with more emphasis due to theoretical models therein directly building on those constructed by Scholes and explicitly tying its experimental observations to the delocalisation of the anion in the fullerene acceptor phase. We have included an extended description of the findings in Ref. 14 in the *Introduction* section as shown above.

3. This paper <https://pubs.acs.org/doi/abs/10.1021/jp5098102> is closely related to the present study and points out the entropic advantage of a large density of CT states for preventing recombination.

We thank the reviewer for this suggestion. We now refer to this paper as a further point to consider following our suggested design rules in the *Discussion* section on page 15:

“Lastly, a small energy gap between the LUMO and higher lying unoccupied orbitals of the acceptor can further support free charge generation by increasing the entropic advantage of CT vs XT states, as suggested in Ref. [71]. In the present system the energy gap between LUMO and LUMO+1 was relatively large (1.8 eV), which is why it was not included in our simulations. However, when this gap is small, the entropic advantage is expected to further enhance the beneficial effects of strong electronic coupling and a small exciton binding energy, in comparison to the results reported in this study.”

Reviewer #2 (Remarks to the Author):

The authors present a detailed theoretical study of nanoscale donor-acceptor interfaces in a prototypical a6T:PDI system, using the X-SH framework to simulate non-adiabatic molecular dynamics. The results indicate that efficient dissociation of excitonic states into free carriers occurs via transiently delocalized hybrid exciton-charge transfer states. Strong electronic couplings between the donor and acceptor molecules promote ‘hot exciton’ dissociation, a similar effect to increasing the dielectric constant of the heterojunction. The authors also discuss the experimental implications of their findings. Additionally, the article introduces an improved X-SH methodology (accounting for off-diagonal disorder) that could benefit future applications to various materials. Given the significance of these findings, I recommend this work for publication in Nature Communications following a revision addressing the comments below. No further review is necessary.

We thank the reviewer for their appreciation of our work and for recommending publication after revision. Below, we provide a point-by-point response to the remaining concerns.

1. The authors consider for their simulations idealized crystalline interfaces between donor and acceptor phases. Indeed, while experimentally it was found that crystalline interfaces work better than amorphous ones for charge separation efficiency, I wonder what the role of defects is. To what extent some typical defects, e.g. faults in the stack of both materials, affect the efficiency of charge transfer mechanisms. These modification can be easily incorporated and tested on the level of X-SH model. This discussion would be an important addition to the paper and point to experimentalists how defect tolerant the charge separation is, for example, for a hot-exciton pathway.

We thank the reviewer for introducing this aspect of stacking defects and its potential effect on charge generation. We have examined this on the level of the X-SH model by introducing an interfacial defect into the PDI acceptor phase, which disrupts its stacking with the a6T donor phase. This allows us to probe the effect of a local defect on exciton dissociation and the dynamics of the CT-states. With an interfacial defect now included in the heterojunction structure, we run an extra X-SH simulation for 10ps where we select the same exciton binding energy and scaling of the electronic coupling as simulation 5 in Table 2, since the CS yield here after 10ps is significant (33.1%). We first refer to these new findings in the *Introduction* in page 4:

“We lastly examine the role played by structural disorder by selecting parameters that led to efficient charge generation in the crystalline a6T:PDI interface, and re-simulating the exciton dissociation process with an interfacial defect introduced to disrupt the donor/acceptor stacking pattern.”

“Additionally, for a parameter combination where the hot path way dominates, the insertion of an interfacial defect into the a6T:PDI heterojunction significantly reduces the rate of exciton dissociation.”

We include the resulting population dynamics and analysis in a new panel in Figure 4, Figure 4d, and the *Results* section (page 12), respectively:

1: New version of Fig. 4 in the manuscript to include the population dynamics of the a6T:PDI heterojunction with the defect, panel (d) shows the comparison of electronic populations of simulation 5 with (dotted) and without (solid) a defect.

“Effect of Interface Defect. So far, we have focussed on the dynamics of free charge photogeneration in a perfectly ordered heterojunction, as a crystalline morphology of the donor/acceptor aggregates has proven beneficial to charge generation efficiency [4] and increased the role played by delocalisation. [26–37] Here, we introduce a prototypical interfacial defect into the PDI acceptor phase (see Fig. S15), to examine its potential effect on exciton dissociation and the dynamics of the CT-states. We focus on the dynamics of a system where free charges are being mostly generated by hot exciton dissociation (simulation 5 in Table 2), as we expect the hot pathway to be most affected by the introduction of a defect. The change in population dynamics upon the introduction of the defect is displayed in Fig. 4d. Exciton dissociation has slowed down markedly, as the disruption in a6T-PDI stacking at the interface has reduced the coupling between the XT and CT states and reduced the density of hybrid XT-CT states, which are necessary for exciton dissociation (see Fig. 3). This results in a significantly lower occupation of CT-states throughout the dynamics. Moreover, a smaller fraction of excitons now dissociate via the hot pathway, whilst a more significant fraction dissociate via the cold pathway that is prone to non-radiative recombination. The interfacial defect stifles the hot pathway by disrupting electron transfer between PDI molecules near the interface. This reduces the density of hybrid XT-CT states with charges delocalised further away from the interface, which are instrumental to hot exciton dissociation (see Fig. 5).”

We then translate these findings into a fourth design rule in the *Discussion* section (page 15):

“4. Structurally ordered donor/acceptor aggregates. The introduction of an interfacial defect that disrupts electron transfer in the acceptor phase near the interface resulted in the cold exciton dissociation pathway gaining significance with respect to the hot pathway, leaving electron-hole pairs more likely to geminately recombine after exciton dissociation. 1.-4. will generally suppress the formation of kinetically trapped ICT states and increase the contribution of hot exciton dissociation.”

We also discuss the creation of the defect in a new section in *Supplementary Information*:

2: Snapshot of a subset of the a6T:PDI heterojunction (Fig. S15) with an interfacial defect introduced into the PDI phase. Coloured molecules are those within the active region in X-SH.

“The a6T:PDI heterojunction with the interfacial defect is shown in Fig. S15. The defect was manually created by taking the final geometry of the NVT equilibration run of the crystalline a6T:PDI heterojunction, deleting the second-most interfacial PDI molecule, and then rotating the interfacial PDI molecule until the ET and XT-CT couplings involving this molecule were reduced by a factor of 4 or more (compared to those reported in Table 1 in the main text). The XT coupling involving the interfacial PDI molecule was only reduced by $\approx 25\%$, since the Coulombic interaction between the S1 transition densities is less sensitive to distance than the orbital overlap. The restraints were then adjusted to reflect the new positions of the upper/lower rings of the interfacial PDI molecule (see *Methods* for information about the application of restraints). The adjusted heterojunction was then propagated in the neutral ground state for 1.1ns in the NVE regime, with the first 100ps allowing the system to equilibrate (in case the intermolecular forces were significantly perturbed by the transformation of the molecule) whilst 1000 geometries were sampled each ps from the remaining 1ns of simulation time. The fluctuation of the temperature around 300 K was very similar to the NVE run of the fully ordered heterojunction.”

2. There are lots of experimental and theoretical studies in the field pointing to the important of vibronic coherence that can fundamentally affect many photoinduced processes and excited state dynamics. The authors use surface hopping dynamics which fundamentally lacks vibrational wave functions, and decoherence corrections are just fixing the non-adiabatic algorithms. The authors need to include a brief discussion on this phenomenon (which to me could be very important for hot exciton dissociation pathway) and how it can alter the emerging picture.

We thank the reviewer for bringing up this interesting point. As he/she correctly notes, surface hopping (SH) is a quantum-classical method and therefore lacks explicitly quantized vibronic wave functions. Nevertheless, SH represents the coupled electron–nuclear wavepacket through an ensemble of classical trajectories, and the nuclear (vibrational) motion is fully considered within this representation. Many nonadiabatic phenomena—such as wavepacket branching, electronic population transfer, and partial delocalization of the electronic wave function—are described with good accuracy by SH across a wide range of systems [T. R. Nelson, *et al. Nat Commun* **9**, 2316 (2018)]. Decoherence is indeed a necessary algorithmic correction whose inclusion, along with others such as the trivial crossing correction and velocity rescaling after a surface hop, is essential to recover the correct dynamics. These aspects have been thoroughly tested in our previous works [Carof, A., *et al. Phys. Chem. Chem. Phys.* **21**, 26368–26386 (2019), Giannini, S., *et al. J. Phys. Chem. Lett.* **9**, 3116–3123 (2018), Carof, A. *et al. J. Chem. Phys.* **147**, 214113 (2017)].

That said, compared to fully quantum-dynamical methods, quantum-classical dynamics may be expected to capture less accurately ultrafast relaxation processes, where the quantum character of high-frequency vibrational modes in some cases affects the wavepacket dynamics (as evidenced by coherent beatings in the populations [Popp, W., Polkehn, M., Binder, R. & Burghardt, I. *J. Phys. Chem. Lett.* **10**, 3326–3332 (2019)] with recurrences of periods shorter than 20 fs). Nevertheless, as we have shown previously for a small model donor-acceptor system through direct comparison of our preliminary X-SH version and fully quantum dynamical simulation at the level of MCTDH (using the same electronic Hamiltonian in both methods), such initial relaxation dynamics are captured with reasonable accuracy by our algorithm, even though the high-frequency modes are treated classically in X-SH [Peng, W. *et al. J. Phys. Chem. Lett.* **13**, 7105–7112 (2022)].

Moreover, we argue that since charge separation occurs on a longer picosecond timescale (for the NFA cells probed by experiments referenced in this study [Bakulin, A.A. *et al.*, *Science*, 335, 1340–1344 (2012); Qian, D. *et al.*, *Nat. Mater.*, **17**, 703–709, 2018]), the vibrational energy in the excited state has been distributed over most vibrational modes. In systems with many degrees of freedom on this longer timescale, the initial vibrational coherences are strongly damped, and what remains is a delocalized (but incoherent) electronic wavefunction. In this regime, we argue that the crucial feature of an algorithm is its ability to distribute excess energy among the various vibrational modes and to capture electronic detailed balance at equilibrium. Our algorithm does so, as demonstrated in our previous work [Giannini, S. *et al. Nat. Commun.* **13**, 2755 (2022); Giannini, S. *et al. Nat. Commun.* **10**, 3843 (2019)], yielding transport properties such as exciton diffusion and charge-carrier mobility in very good agreement with experimental results.

We have included a paragraph on the implications of using quantum-classical dynamics in simulating charge separation in the *Discussion* section (page 13) of the main text:

“Besides dKMC, fully quantum-dynamical methods have been employed to study charge separation in simplified donor–acceptor interface models. Previous studies have shown that the quantum nature of high-frequency vibrational modes can, in some cases, influence the wavepacket dynamics, as evidenced by coherent beatings in the populations [60]. These coherent oscillations can affect relaxation processes and charge separation on ultrafast timescales. From our previous comparisons with fully quantum-dynamical simulations (using MCTDH for the same electronic Hamiltonian), we found that such initial dynamics are captured with reasonable accuracy by X-SH, even though the high-frequency modes are treated

classically. An important benefit of our approach over a fully quantum-mechanical treatment, beyond the significant computational efficiency, is that our scheme correctly recovers the long-time detailed balance, a fundamental requirement for processes occurring on extended timescales where essentially all vibrational modes are in thermal equilibrium and vibronic coherences are significantly damped [48].”

3. Cold exciton dissociation. Is it correct to assume that the simulations are limited by 10ps of simulations time. Within this timescale, there was no observed dissociation of the iCT states. The subsequent dynamics (i.e. “charge separation or recombination of iCT state”) involves longer timescales and this dynamics is not included in the kinetic models presented. Perhaps some clarifying remarks would be helpful here.

We thank the reviewer for suggesting clarification of this point. Each X-SH simulation in this study was indeed limited to 10ps due to its computational expense, so we focused on comparing the X-SH population dynamics to the first 10ps of the kinetic model. For X-SH simulations where cold exciton dissociation dominates (e.g. Table 2, simulation 1), we did not observe any dissociation of the iCT states. This indicates that, for the parameter regimes herein, X-SH predicts that the iCT states either recombine or dissociate on timescales longer than 10ps.

We have introduced an additional statement when explaining the observed cold pathway in the *Results* section (page 8), to clarify that we investigate charge generation within the first 10ps due to the length of the X-SH simulations:

“The subsequent dynamics, that is charge separation or recombination of the iCT state, involves longer time scales. This dynamics is not included in the present simulations as these longer timescales cannot be presently accessed. The mechanism described above is referred to as “cold exciton dissociation” because all excitons relax to the lowest excited (“cold”) state of the system, i.e. iCT state, before they acquire enough thermal energy to separate to free charges or recombine to the ground state.”

4. Hot exciton dissociation. I would question validity of the statement “a small increase in dielectric constant from 3.5 to 5 (Table 2, simulation 4) results in a decrease of exciton binding energy from 260 to 10 meV.” This is a huge claim and I do not think that just referring to Eq. 24 provides justice. Exciton binding energy is a delicate quantity to calculate. I would request the authors to substantiate their claims. Experimental evidence is indirect (ref. 61).

We thank the reviewer for recommending clarification of this point. We agree that the calculation of very accurate values of the exciton binding energy is a non-trivial task. First, we would like to note that Eq. 24 is a widely accepted definition for exciton binding energy [Savoie, B.M. *et al. Acc. Chem. Res.* **47**, 3385–3394 (2014); Yufan, Z. *et al. Adv. Energy Sustainability Res.* **3**, 2100184 (2022)]. Second, for the purpose of this work it is sufficient to calculate the terms in Eq. 24 with some common approximations, as outlined in the Methods section and section S3 of the SI. The exciton binding energies quoted in the main text are valid within these approximations.

We have introduced additional comments in the *Results* section of the main text (page 9) highlighted by the reviewer, to clarify the approximations we have made upon calculating the exciton binding energy of the a6T:PDI heterojunction:

“Alternatively, a small increase in dielectric constant from 3.5 to 5 (Table 2, simulation 4) shifts the niCT and CS bands down in energy to align with the XT band (Fig. 2c). Adopting the common expression for exciton binding energy [53] (Eq. 24) and approximating the formation energy of free charges from the iCT state by a screened Coulomb potential, we find that the increase in dielectric constant corresponds to a lowering of the exciton binding energy from 260 to 10 meV (however, we note that this is an approximation as the electron-hole interaction energy at short range may not exactly follow a screened Coulomb interaction). In both cases, with an increase of the electronic coupling or dielectric constant, the resulting close energetic alignment also increases the density of hybrid XT-CT states.”

5. Regarding the abstract statement: “Our results highlight the importance of electronic couplings, an often overlooked property, for opto-electronic charge generation.” I believe that this is not true and tuning of electronic coupling has been an active subject of the discussion in the community for decades across both theoretical and experimental studies. Perhaps this statement and similar tone in the main body of the article need to be softened.

We agree the effect of electronic coupling on optoelectronic charge generation has been discussed before (e.g. by Balzer *et al.* in Ref. 34). Here, we intended to highlight that this important materials property has received much less attention for optimisation of materials for organic solar cells than it has for organic electronics. In response to the reviewers suggestion we have softened the wording and changed the sentence in the abstract to:

“Thus, our results highlight the importance of the electronic coupling and dielectric constant for opto-electronic charge generation.”

6. The present approach considers singlet states only. However, I would argue that weakly bound triplet charge transfer states are iso-energetic with their singlet counterparts and can be easily form even in the absence of strong spin-orbit coupling. Their presence and participation in the process can add additional entropic driving force further improving charge separation. I would suggest to add discussion of this scenario.

We thank the reviewer for this interesting point. We have added a discussion on triplet formation in the Discussion section (page 14):

“As mentioned earlier, we only considered singlet states in this work, disregarding any potential formation of low-lying triplet states. Gillett *et al.* provided a comprehensive experimental assessment of the role of triplet CT states in low-offset and NFA-based organic solar cells. [64] They can be close in energy to their singlet counterparts because the exchange energy decreases rapidly with increasing electron-hole separation. [65] Triplet CT states may form via a geminate spin-mixing mechanism from singlet CT states on nanosecond timescales; however, Gillett *et al.* found no experimental evidence for this geminate pathway in the NFA blends they examined. Instead, they identified a dominant non-geminate mechanism, in which the recombination of free charges generates triplet CT excitons that undergo fast back charge transfer to Frenkel exciton triplets. However, the non-geminate recombination pathway requiring free charge carriers happens on much longer time scales than the current X-SH simulations can access. Yet, depending on the material under consideration, triplet formation may occur faster, in which case it provides additional entropic driving force for free charge generation [66,67].”

7. Code availability statement says: “The custom codes used for this study are available from the corresponding authors upon request.” It would be really helpful to clarify releases of the

code for public, for example, if any of the standard version of X-SH are available and what are the plans for realizing the improved version. This is an important advance of this study.

We agree with the reviewer that a public release of the X-SH code is beneficial for the advancement of this study. The code utilised in the X-SH simulations presented herein is incorporated into the latest version of our in-house modified CP2K-V4.1 package. This is a development version, and the latest fully tested version of the code is publicly available on <https://github.com/blumberger/flavoured-cptk-public-copy.git>. We have also amended our Code Availability statement accordingly:

“The latest tested version of the custom code used for this study is publicly available on <https://github.com/blumberger/flavoured-cptk-public-copy.git>.”

Reviewer #2 (Remarks on code availability):

I do not think that X-SH code used for this study was released, see my point #7

Please see the answer above.

Additional changes:

We have slightly amended two sentences to make better use of supplementary figures in supporting the conclusions drawn in the main text (page 11):

“This novel state is a superposition of delocalised XT, iCT and niCT states. In such states, the excitons and charges are delocalised to a greater extent (see Fig. S6) and can be centred further from the donor/acceptor interface (Fig. S8).”

“A remarkable feature of the hot exciton dissociation mechanism is that a significant proportion of excitons no longer dissociate at the interface, as traditional models of excitonic solar cells would assume[2], but several lattice spacings / nanometres away from it, owing to the delocalised nature of hybrid XT-CT states in this regime (see Figs. S4 and S10, using the center of the wavefunction as a distance measure).”